# The Beneficial Effect of Mitochondrial Transfer Therapy in 5XFAD Mice via Liver–Serum–Brain Response

**DOI:** 10.3390/cells12071006

**Published:** 2023-03-24

**Authors:** Sahar Sweetat, Keren Nitzan, Nir Suissa, Yael Haimovich, Michal Lichtenstein, Samar Zabit, Sandrine Benhamron, Karameh Akarieh, Kumudesh Mishra, Dinorah Barasch, Ann Saada, Tamar Ziv, Or Kakhlon, Haya Lorberboum-Galski, Hanna Rosenmann

**Affiliations:** 1Department of Neurology, The Agnes Ginges Center for Human Neurogenetics, Hadassah Hebrew University Medical Center, Jerusalem 9112001, Israel; 2Faculty of Medicine, Hebrew University of Jerusalem, Jerusalem 9112102, Israel; 3The Smoler Protein Research Center, Technion Israel Institute of Technology, Haifa 3200003, Israel; 4Department of Biochemistry and Molecular Biology, Institute for Medical Research Israel-Canada (IMRIC), Faculty of Medicine, Hebrew University of Jerusalem, Jerusalem 9112102, Israel; 5Mass Spectrometry Unit, Institute for Drug Research, School of Pharmacy, Hebrew University of Jerusalem, Jerusalem 9112102, Israel; 6Department of Genetic and Metabolic Diseases, Hadassah Hebrew University Medical Center, Jerusalem 9112001, Israel

**Keywords:** mitochondria, Alzheimer’s disease, mitochondrial transfer, 5XFAD, amyloid, cognition

## Abstract

We recently reported the benefit of the IV transferring of active exogenous mitochondria in a short-term pharmacological AD (Alzheimer’s disease) model. We have now explored the efficacy of mitochondrial transfer in 5XFAD transgenic mice, aiming to explore the underlying mechanism by which the IV-injected mitochondria affect the diseased brain. Mitochondrial transfer in 5XFAD ameliorated cognitive impairment, amyloid burden, and mitochondrial dysfunction. Exogenously injected mitochondria were detected in the liver but not in the brain. We detected alterations in brain proteome, implicating synapse-related processes, ubiquitination/proteasome-related processes, phagocytosis, and mitochondria-related factors, which may lead to the amelioration of disease. These changes were accompanied by proteome/metabolome alterations in the liver, including pathways of glucose, glutathione, amino acids, biogenic amines, and sphingolipids. Altered liver metabolites were also detected in the serum of the treated mice, particularly metabolites that are known to affect neurodegenerative processes, such as carnosine, putrescine, C24:1-OH sphingomyelin, and amino acids, which serve as neurotransmitters or their precursors. Our results suggest that the beneficial effect of mitochondrial transfer in the 5XFAD mice is mediated by metabolic signaling from the liver via the serum to the brain, where it induces protective effects. The high efficacy of the mitochondrial transfer may offer a novel AD therapy.

## 1. Introduction

Alzheimer’s disease (AD) is a neurodegenerative disease affecting many cellular pathways, including protein aggregation, deterioration of neurotransmission, mitochondrial dysfunction, oxidative stress, and neuroinflammation, leading ultimately to neuronal death [1,2]. Functional mitochondria are critical for the normal activity of all cells, with neurons highly dependent on mitochondrial function because of their limited glycolytic capacity [3,4]. Mitochondria impairment is involved in neurodegeneration-related processes, such as oxidative stress (via reactive oxygen species (ROS) generation, leading to oxidative degeneration of macromolecules [5,6]), apoptosis [7], iron metabolism [8], energy imbalance [9], and interaction with Aβ and phosphorylated tau-protein [10,11], together with our recent study showing that mitochondrial toxin induced AD/tauopathy brain pathology [12]. Therefore, targeting mitochondrial impairments may provide a preferred strategy. However, since mitochondria affect many pathways, targeting them is not a simple goal. Furthermore, targeting one specific mitochondrial defect may be ineffective, since the defect may be due to other processes. Thus, mitochondrial therapy is challenging and difficult to regulate precisely. To overcome this limitation, we suggest an AD therapy based on the transfer of exogenous active mitochondria as intact cell organelles. This allows a broad range of activities with accurate and physiological regulation. Mitochondrial transfer is a process by which isolated mitochondria can be incorporated in vitro into cells by simple incubation, first reported in 1982 by Clark [13], and their functionality inside the recipient cells has been demonstrated [14,15]. Mitochondria enter the cells via macropinocytosis or endocytosis, as shown by others and by us [16,17,18]. Transfer/transplantation of isolated mitochondria in experimental animal models, mostly locally but also systemically delivered, showed beneficial effects, as reported in injury, including ischemia [19,20,21,22,23,24,25,26,27], with a few studies showing efficacy also in Parkinson’s disease and schizophrenia symptoms [28,29]. Interestingly, a very low level of mitochondria internalization in vivo into the target organ was reported [19,21,30,31,32], raising important unsolved questions regarding the mechanism involved in mitochondrial transfer therapy.

Recently, for the first time as a treatment for a cognitive disease, we reported a robust beneficial effect of intravenous (IV) mitochondrial transfer in a short-term pharmacological model of AD (intracerebroventricular (ICV)-injected amyloid-beta, Aβ) as proof of concept for AD and dementia. Importantly, this treatment showed a high safety profile (normal general health and internal organs, with no immune response). IV-injected labeled mitochondria were not detected in the brain, while, interestingly, they were detected in the liver [33]. Here, we report the outcomes of mitochondrial transfer therapy in a chronic long-term AD model: 5XFAD Tg mice. We investigated mechanistic aspects of the brain and liver response to this therapy, aiming to shed light on the beneficial effect of the treatment, which is not necessarily mediated by mitochondria internalization into the brain. We show that mitochondrial transfer in the 5XFAD mice ameliorated cognitive impairments, reduced neuronal damage and amyloid burden, and improved mitochondrial activities. Hippocampus proteomics showed alteration in proteins that may lead to disease amelioration, such as changes in synapse-related processes, GABA-regulation and histone deacetylase, ubiquitination/proteasome-related processes, regulation of Fc signaling/phagocytosis, and mitochondrial factors. Liver proteomics and metabolomics showed the involvement of a broad metabolic with mitochondrial response. Altered liver metabolites were also detected in the serum of the treated mice, including metabolites that affect neurodegenerative processes, such as carnosine, putrescine, and C24:1-OH sphingomyelin, and amino acids, which serve as neurotransmitters or their precursors. These results may indicate that the beneficial effect of mitochondrial transfer on cognition, brain pathology, and mitochondrial function in 5XFAD mice is mediated by metabolic signaling from the liver via the serum to the brain. 

## 2. Materials and Methods 

### 2.1. Mitochondria Isolation 

We isolated mitochondria from HeLa DsRed (Discosoma sp. red fluorescent protein) 2-mito cells, previously transfected by us with the plasmid DsRed2-Mito (Clontech, Mountain View, CA, USA) [18]. (Appendix A).

### 2.2. Mice

Hemizygous 5XFAD transgenic (Tg) males that co-overexpress FAD mutant forms of human APP and PS1 mice were crossed with C57Bl/6J (or C57BL/6JRCCHSD) females. Offspring were genotyped as described [34]. Negative offspring served as non-AD mice. All experiments were approved by the Institutional-Ethics-Committee of The Hebrew-University of Jerusalem (Approval Number: MD-15-14651-5).

### 2.3. Mitochondrial Transfer Treatment

Transfer of mitochondria (IV tail vein injection: 200 μg mitochondria/mouse (a dose which we used in our recent study in the pharmacological AD model [33]), given every 2 weeks) was conducted in 3 experiments: 1. AD mice on C57Bl/6J background (three males, two females) received four injections from the age of 6 months, while untreated AD mice (three males, one female) and non-AD mice (six males, five females) received mitochondrial buffer only. 2. AD mice on C57BL/6JRCCHSD (nine males) received three injections from the age of 6 months, while untreated AD mice (eight males) and non-AD mice (eight males) received buffer only. 3. AD mice on C57BL/6JRCCHSD (seven females) received two injections from the age of 12 months, while untreated AD mice (five females) and non-AD mice (eight females) received buffer only. 

In addition, to test whether the injected mitochondria arrive at the brain and/or liver, these organs were collected from 6–8 month-old AD mice 2 h following the DsRed-mitochondria or buffer-only injection (*n* = 2/group).

### 2.4. Behavioral Studies

A few days (3–7 days) following the last mitochondrial transfer, cognitive tests were performed: Y-maze, Open-Field Habituation, Novel Object Recognition, and T-maze, using protocols previously reported by us [33,35,36] (Appendix A).

### 2.5. Histopathological Studies

After finalizing the behavioral tests (about 2 weeks after last mitochondrial injection) mice were anesthetized with a lethal dose of pentobarbital and perfused via the ascending aorta with ice-cold PBS. Brains and livers were removed. One-half of the brain was fixed in 4% paraformaldehyde for histological studies, and the other half was stored at −80 °C for biochemical and proteomics studies. Similarly, part of the liver was fixed for histological studies, and another part was stored at −80 °C for biochemical and proteomics/metabolomics studies.

Brain sections were stained for amyloid burden [6E10 (Biolegend, CA, USA), thioflavin S (Sigma Aldrich)] and neuronal loss [anti-neuronal nuclei, NeuN (Millipore Burlington, MA, USA), Fluoro-Jade (Sigma Aldrich)]. Brain and liver sections were stained for the presence of the DsRed-2-labeled mitochondria [anti-red fluorescent protein, RFP (MBL Co Ltd., Japan)]. Analysis was performed in a blinded manner, analyzing three sections per animal. In the brain, NeuN-, FJD-, and thioflavin-stained cells were counted manually using the Image-J 1.52a software. For 6E10 analysis, signal intensity was integrated to measure fluorescence signal density strength with the Nis elements software Ver. 4.0 (density per binary area was measured). The same region of interest (ROI) was selected in the cortex of each animal. The presence of RFP-stained cells was tested in various regions in the brain and in the liver sections. The sections were imaged by fluorescent microscopy (X20, Nikon-TL, or Nikon confocal A1R microscope). We used protocols previously reported by us [33,35,36] (Appendix A).

### 2.6. Imaging Analysis

Brains and livers were subjected to imaging analysis to detect the DsRed signal by an in vivo imaging system (IVIS) (Appendix A). 

### 2.7. Mitochondrial Enzymatic Assays 

Enzymatic assays were performed in homogenates of cortex and liver samples. The activity of the Cytochrome c oxidase (COX), which is the 4th complex of the electron transfer chain, and of the Citrate synthase (CS), which is the ubiquitous mitochondrial matrix Krebs-cycle enzyme (used as a mitochondrial marker enzyme), were tested using our previously reported protocols [33]. Enzymatic activities were expressed as a ratio normalized to CS activity.

### 2.8. Proteomic Studies

Proteins were extracted from hippocampal samples, separated to soluble and aggregate fractions, trypsinized, and analyzed by liquid chromatography-tandem mass spectrometry (LC-MSMS) using the Q-Exactive HF (Thermo Fisher Scientific). The mass spectrometry data were analyzed using the MaxQuant software 1.5.2.8 for peak picking and identification using the Andromeda search engine. The intensities of the proteins between the samples were quantified by Label-free quantitation (LFQ). Differential proteins were determined based on a statistical test of proteins’ normalized intensities between the experimental groups. Altered proteins were divided into the following categories,: Tg Effect, Treatment Rescued, Treatment Effect (similar to the categories described by Koren et al.) [37]. Proteins that were extracted from liver samples were analyzed similarly (Appendix A).

### 2.9. Metabolomic Studies 

To capture a broad spectrum of metabolites in the liver and serum, we used the AbsoluteIDQ^®^ p180 kit (Biocrates Life Sciences AG, Innsbruck, Austria) similarly to the protocol reported by us previously [38], using the MetaboAnalyst 5.0 [39] (Appendix A).

### 2.10. Statistics

The data are presented as mean ± SEM. Data were analyzed using one-way ANOVA, further analyzed using LSD or Tukey post-hoc test, and when mentioned, unpaired *t*-test. Statistical analysis was performed using GraphPad Prism 8. We used the term “trend” aiming to describe results with weak evidence, similar to the definition in VSNI (data science software and experimental design software for biosciences, https://vsni.co.uk/blogs/what-is-a-p-value; accessed on 14 February 2023). Statistical significance was accepted at *p* < 0.05 (*) and trends at *p* < 0.1 (^) [40]. Statistical analysis of the proteomics and of metabolomics are described in the “Proteomic Analysis” and in “Metabolomic Analysis” sections, respectively.

## 3. Results

### 3.1. Amelioration of Cognitive Deficits in AD Mice Treated with Mitochondrial Transfer

AD mice treated with mitochondria were tested for their cognitive performance compared to untreated AD and non-AD mice. AD mice (5XFAD on C57BL/6J background) received four IV injections of the isolated mitochondria (every 2 weeks) starting at 6 months of age. Treated AD mice showed better cognitive performance compared to untreated mice, approaching the performance of non-AD mice. Improved performance in the mitochondria-treated AD mice was detected in the Open-Field Habituation test: there was a significantly lower distance moved on Day 2 compared to Day 1 in the non-AD mice (*t*-test, *p* < 0.001) and in the treated AD mice (*p* < 0.001), indicating intact recollection of the maze, while the untreated AD mice did not show a significant difference between the days (Figure 1A). In the Y-maze test, one-way ANOVA showed a trend toward difference between the groups [f(2, 16) = 3.22, *p* = 0.06]. LSD post-hoc analysis showed significantly worse performance in AD mice relative to the non-AD group (*p* = 0.02) and a trend toward better performance in treated AD mice relative to untreated AD mice (*p* = 0.09), with no difference between treated AD mice and non-AD mice (Figure 1B). When testing AD mice (on C57BL/6JRCCHSD background), which received three IV injections of the isolated mitochondria starting at 6 months of age, we detected better cognitive performance compared to untreated AD mice, performing similarly to non-AD mice, as follows: in the T-maze test, one-way ANOVA showed a significant difference between the groups [f(2, 22) = 3.84, *p* = 0.037]. LSD post-hoc analysis showed significantly better performance in treated AD mice compared to untreated AD mice (*p* = 0.01) and a trend toward worse performance in AD mice relative to the non-AD group (*p* = 0.1), with no difference between treated AD mice and non-AD mice (Figure 1C). Additionally, in the Novel Object Recognition test, better cognitive performance in treated AD mice compared to untreated AD mice was noted, as follows: one-way ANOVA showed a significant difference between the groups [f(2, 22) = 3.76, *p* = 0.039]. LSD post-hoc analysis showed significantly worse performance in AD mice relative to the non-AD group (*p* = 0.01), with no difference between treated AD-mice and non-AD mice. (Figure 1D). Similar results were recorded in older mice, even with fewer mitochondria injections. In AD mice (5XFAD on C57BL/6JRCCHSD background), which received two IV injections of the isolated mitochondria starting at 12 months of age, we detected better cognitive performance compared to untreated AD mice, as follows: in the T-maze test, one-way ANOVA showed a significant difference between the groups [f(2, 16) = 3.477, *p* = 0.05]. LSD post-hoc analysis showed significantly worse performance in AD mice relative to the non-AD group (*p* = 0.01), and a trend toward worse performance in AD mice relative to the treated AD group (*p* = 0.09), with no difference between treated AD mice and non-AD mice. (Figure 1E). Similarly, in the Novel Object Recognition test, one-way ANOVA showed a significant difference between the groups [f(2, 16) = 4.463, *p* = 0.029]. Tukey post-hoc analysis showed significantly worse performance in the AD mice compared to the non-AD mice (*p* = 0.003), with no difference between the treated AD mice and non-AD mice, suggesting better cognitive performance in the treated mice (Figure 1F). 

### 3.2. Reduced Neuronal Damage and Amyloid Burden in the Brain of AD Mice Treated with Mitochondrial Transfer 

NeuN immunohistological staining of neurons in the cortex revealed that while there was a significantly lower neuronal count (higher neuronal loss) in the AD mice than in the non-AD mice, the mitochondria-treated AD mice had a higher neuronal count than untreated AD mice, as follows: one-way ANOVA showed a significant difference between the groups [f(2, 12) = 12.59, *p* = 0.001]. LSD post-hoc analysis showed a significantly lower neuronal count in AD vs. non-AD mice (*p* = 0.0003), with a higher count in mitochondria-treated vs. untreated AD mice (*p* = 0.04). Similar results were obtained by FJC staining for degenerative neurons. Degenerative neurons were detected in the cortex of the AD mice, with a decreasing trend in the mitochondria-treated AD mice (*p* = 0.1), and were hardly seen in the cortex of the non-AD mice (Figure 2A–C). 

Staining for amyloid burden was conducted using 6E10 Ab for amyloid and thioflavin for the amyloid plaques. A decrease in 6E10 staining was detected in the mitochondria-treated AD mice compared to the untreated AD mice (*t*-test, *p* = 0.009). These results were further supported by the trend toward decrease in the thioflavin staining for plaques (*p* = 0.06). Staining of both 6E10 and thioflavin was hardly detected in the non-AD mice (Figure 2D–F). 

### 3.3. Increased Mitochondrial Enzymatic Activity in the Brain and Liver of AD Mice Treated with Mitochondrial Transfer 

We investigated whether mitochondrial respiratory chain activity in both the brain and the liver would respond to the mitochondria therapy. To this end, we tested the effect of IV mitochondrial transfer on the mitochondrial COX enzymatic activity in the cortex and the liver of the treated AD, untreated AD, and non-AD mice. 

In mice treated with mitochondria from the age of 6 months (four IV injections), we detected an increase in the mitochondrial activity expressed as COX/CS ratio (COX normalized to CS activity) in the cortex and the liver in the mitochondria-treated AD mice compared to the untreated AD mice, reaching, and even exceeding, the activity of the non-AD mice, as follows: One-way ANOVA showed a significant difference in the brain between the groups [f(2, 11) = 8.064, *p* = 0.007]. LSD post-hoc analysis showed a significantly lower ratio in AD vs. non-AD mice (*p* = 0.002), with a higher ratio in the mitochondria-treated vs. untreated AD mice (*p* = 0.018), reaching the ratio of the non-AD mice (Figure 3A). A beneficial effect of the mitochondrial transfer was also evident in the liver: a higher COX/CS ratio was detected in the treated AD mice relative to the untreated AD mice (*t*-test, *p* = 0.004), reaching an even higher value than that in the liver of the non-AD mice, with a trend showing a reduced ratio in the AD relative to non-AD mice (*p* = 0.1) (Figure 3B). A beneficial effect was also detected in the cortex of aged AD mice (12 months, two injections). The treated AD mice showed a higher COX/CS ratio relative to the untreated AD mice (*t*-test, *p* = 0.028), reaching the ratio of the non-AD mice and even exceeding it, with a trend showing a reduced ratio in the AD mice relative to non-AD mice (*p* = 0.15) (Figure 3C). A similar effect was also noted in the liver: one-way ANOVA showed a trend toward difference in the liver between the groups [f(2, 16) = 3.245, *p* = 0.06]. LSD post-hoc analysis showed a significantly higher COX/CS ratio in the treated compared to the untreated AD mice (*p* = 0.044), with a reduced ratio in the AD relative to non-AD mice (*p* = 0.033) (Figure 3D). 

### 3.4. IV-Injected Mitochondria Are Detected in the Liver but Not in Brain

We investigated whether the DsRed signal of the IV-injected DsRed-labeled mitochondria reached the brain and the liver. IVIS or anti-RFP immunostaining revealed a signal in the liver of AD mice following the mitochondria injection but not in their brain (in Figure 3E,F). This result suggests that the beneficial effect of the IV-injected mitochondria is not mediated by a direct entry of the mitochondria into the brain and that a liver response is taking place. 

### 3.5. Proteome Alterations in the Brain of AD Mice Treated with Mitochondrial Transfer 

Comparative proteomic analysis was performed to identify proteins affected by the transgenicity and the treatment in the AD mice receiving four IV injections of mitochondria from the age of 6 months compared to untreated AD mice and non-AD mice. Eighty-three proteins were identified in the hippocampal homogenate proteomes as significantly altered across all groups (non-AD, AD, and treated AD), as shown in the hierarchical cluster heatmap (Figure 4). We next analyzed the altered proteins, comparing separately the AD vs. non-AD mice, the AD vs. treated AD mice, and the treated AD vs. non-AD mice. We then identified the proteins rescued by the treatment: proteins increased (or decreased) in the AD mice relative to non-AD mice (proteins of “Tg effect”) and decreased (or increased) in the treated AD mice, similar to the levels in the non-AD mice (meaning that the treatment corrected/lessened the AD vs. non-AD difference) (termed “Treatment Rescued” proteins). These included Clic 1, Apod, mitochondrial ribosomal proteins (Mrpl38, Mrps6), and Aβ 1–40 and 1–42 peptides. Proteins with an AD vs. treated AD difference and without an AD vs. non-AD difference were termed “Treatment Effect” proteins, as the treatment affected the protein level unrelated to a Tg Effect. These included Mib1, Mib2, and Gabra5 (Appendix A).

The mitochondria treatment effect in the AD mice was notably evident when analyzing the altered proteins in the aggregated fraction of the hippocampus homogenates separately. Among the altered proteins were Aβ 1–40 and 1–42, Psma3, and Ggt7 (Treatment Rescued) and Mib2, Pja1, and mitochondrial proteins (Ntfdh, Ndufa11, Cox6a1, Coa3, Mief1, Myg1, Shmt2, Slc6a4) (Treatment Effect proteins) (Appendix A). While Aβ 1–40 and 1–42 peptides were reduced by the mitochondrial transfer treatment, there was no effect on the human APP (hAPP, the transgenic gene) (FC 0.75, *p* = 0.47) (Appendix A). 

Annotation analysis, using the Kyoto Encyclopedia of Genes and Genomes (KEGG) and Gene Ontology Term Enrichment (GOTERM) database for testing the components and processes enriched in the hippocampus in response to treatment, revealed the involvement of ubiquitination, RNA processing, ion/chloride transport, synaptic transmission/post-synaptic regulation/GABA signaling, blood coagulation, oxidative phosphorylation, and regulation of mitochondrial translation and fission, particularly in the aggregated fraction.

The Gene Ontology enRIchment anaLysis analysis and visuaLizAtion tool (Gorilla) algorithm revealed a similar enrichment, including ubiquitin ligase complexes, histone deacetylase (HDAC) complex, GABA receptor complex, post-synaptic membrane, regulation of mRNA metabolic process, blood microparticles, and response to stress (Figure 5A,B). In the aggregated fraction small molecule metabolic process, the regulation of the Fc-receptor-mediated stimulatory signaling pathway, carboxylic acid metabolic process, positive regulation of phagocytosis, and others was enriched (Figure 6A,B). 

Post-translational modification (PTM) analysis of the proteins, particularly phosphorylation, revealed alteration in MAP2, Stat6, and others (Treatment Rescued), and in GSK3a, Bsn, Padha1, neurofilament polypeptides (heavy, Nefh; medium, Nefm), and others (Treatment Effect) (Appendix A). Analyzing the phosphorylation amino-acid motifs of the differential phospho-peptides between the AD and treated AD groups revealed many peptides with the SP motif that can point towards map kinase involvement (Figure 7A). STRING, the protein–protein Interaction Networks Functional Enrichment Analysis for phosphorylation modifications, showed the high number of synapse-related proteins altered in response to the mitochondrial transfer treatment (Figure 7B). 

These proteomic studies of the hippocampus pointed to alterations following treatment in proteins, which may lead to amelioration of cognitive impairment, amyloid burden, and mitochondrial dysfunction. 

### 3.6. Proteome Alterations in the Liver of AD Mice Treated with Mitochondrial Transfer 

We conducted a similar proteomic analysis on the liver of the treated AD mice to further shed light on the liver response to the treatment. Thirty-two proteins were identified in the liver proteome as significantly altered across all group comparisons (Figure 8A). When analyzing the altered proteins by comparing separately the AD vs. non-AD mice, the AD vs. treated AD mice, and the treated AD vs. non-AD mice, we identified Treatment Rescued proteins, such as R3hcc1, Pik3r1 and Gstp3, and Treatment Effect proteins (without a Tg effect), such as Igfbp2 and Glb1 (Appendix A).

Annotation analysis revealed the enrichment of the peroxisome proliferator-activated receptor (PPAR) signaling (a signaling of PPAR transcription factors that cross-talk with insulin signaling and modulation of insulin sensitivity [41]; it is also connected with the altered pathway of the AMP-activated protein kinase (AMPK) signaling) [42], other pathways involved in glucose metabolism (phosphoinositide 3-kinase (PI3K)-Akt signaling, glucagon signaling, insulin resistance, and gluconeogeneis) [43,44,45], and also vascular endothelial growth factor (VEGF) signaling, regulation of lipolysis, apoptotic signaling, autophagy, protein ubiquitination, glutathione metabolic process (KEGG/GOTERM database) (Figure 8B). Also affected are mitochondrial respiratory chain complex I, lysosome, muscle contraction, cellular amino acid metabolic process, small molecule metabolic/biosynthetic process, lipid biosynthetic processes, response to stress and immune response, and others (Gorilla algorithm) (Figure 8C). 

### 3.7. Metabolome Alterations in the Liver of AD Mice Treated with Mitochondrial Transfer 

To further address the possibility that a liver-related response to the mitochondria IV injection is taking place, we performed targeted metabolomics on the liver to identify metabolites and their pathways affected by the transgenicity and the treatment. Partial least square discriminant analysis (PLS-DA, “supervised PCA”) obtained from the three groups (non-AD, AD, and treated AD) showed separation of the samples, with the AD mice and the non-AD mice showing the biggest difference, while the treated AD group separated from the AD mice (Figure 9A). Eight metabolites were identified as significantly altered across all group comparisons, presented in the hierarchical cluster heatmap (ANOVA, *p* < 0.05) as an average value for each group, with most of them showing Tg effect (Figure 9B). Metabolites such as carnosine and C24:1-OH sphingomyelin showed Treatment Rescued effect, while metabolites such as Asn and hexanolcarnitine showed Treatment Effect. 

The levels of the altered metabolites, when comparing separately the AD vs. non-AD, treated AD vs. non-AD, and AD vs. treated AD mice, showed Treatment Rescued metabolites (hydroxytetradecenoylcarnitine, hexadecenoylcarnitine, C24:1-OH sphingomyelin, SM(d18:0/20:2), and carnosine) as well as Treatment Effect metabolites (hexanolycarnitine, Ala, Asn, Asp, and putrescine) (Appendix A). Complementary support for the involvement of these altered metabolites, particularly the amino acids, is demonstrated in the VIP (measure of a variable’s importance) score plots, presenting the contribution of the 25 metabolites (out of the 188 tested) with the highest impact on the difference between the tested groups (Figure 9C). Referring to the metabolites’ effect on a relativity basis (Appendix A), the VIP score comparisons revealed that hexose, taurine, glutamine, and Glu have a Treatment Rescued effect, while Asn, Asp, and Ala showed a Treatment Effect. Alteration in Gly contribution may suggest some adaptation in the AD compared to non-AD mice by an increase in Gly contribution that was enhanced in the treated AD mice. 

We next tested the correlations of the metabolites with the transitions (differences) between the groups. Figure 9D presents the top 25 metabolites correlated with the transitions. Among the positively correlated metabolites with the non-AD vs. AD transition are carnitine and acetylcarnitine, which are associated with the beta-oxidation of fatty acids. Most of the metabolites showing a positive correlation in the non-AD vs. AD groups do not correlate in the non-AD vs. treated AD, suggesting that the mitochondria treatment reduced the Tg difference (Treatment Rescued effect). This is further supported by the finding that some of these positively correlated non-AD vs. AD metabolites showed negative correlation in the AD vs. treated AD, i.e., had an opposite effect, therefore rescuing the AD Tg deficit relative to non-AD, such as carnosine, sphingomyelin, hydroxytetradecenocarnitine, and nexadecenolcarnitine. The AD vs. treated AD difference showed correlated metabolites not detected in the non-AD vs. AD comparison, suggesting a Treatment Effect (not related to transgenicity). Among these metabolites are various amino acids, including Ala, Asn, and Asp (positively correlated) and hexanolcarnitine (negatively correlated) (Figure 9D), in accord with the comparison of the levels of metabolites among the tested groups (Appendix A).

Metabolomic Pathway Analysis, using the the Small Molecule Pathway Database (SMPDB) pathway libraries as references, for identifying the metabolic pathways enriched by the mitochondria treatment is presented in Figure 9E. The 5–6 top enriched sets in the treated-AD vs. AD mice included glucose-alanine cycle ammonia, tryptophan metabolism, selenoamino acid metabolism, glutathione metabolism, alanine metabolism, glycine, and serine metabolism. The top enriched pathways of the AD vs. non-AD groups showed mostly beta-oxidation pathways (Appendix A). 

Taken together, the metabolomic analyses revealed that the liver response to the mitochondrial therapy involves alterations in the levels of amino acids and biogenic amines, as well as of sphingolipids and acylcarnitines.

### 3.8. Integrative Analysis of Proteome and Metabolome Alterations in the Liver

Integrating the changes discovered in the metabolomic and the proteomic profiles of the liver in response to mitochondrial transfer therapy (similar to the report testing the interaction of lipids with the activity of enzyme-protein in the liver [46]) revealed alterations in amino acid metabolism, glutathione metabolism, glucose metabolism, sphingolipid metabolism, and acylcarnitines (within mitochondria, as the acylcarnitines are derivatives of long-chain fatty acids, which are required for the transport of these fatty acids into mitochondria for beta-oxidation) [47,48] (Table 1). A representative example is that while metabolomics predicts activation of the gluconeogenetic pathway “glucose anlanine cycle”, proteomics also demonstrates an increase in gluconeogenesis proteins. The metabolomics prediction included altered sphingolipid metabolism, which is also seen in the proteomics as alterations in the associated proteins Pik3r1 and Glb1 [49,50]. These results show accordance between liver metabolomic and proteomic alterations in response to the mitochondrial transfer therapy, further supporting the broad metabolic reaction of the liver.

### 3.9. Metabolome Alterations in the Serum of AD Mice Treated with Mitochondrial Transfer 

To address the possibility that the liver-related metabolic response to mitochondria IV injection may project to the brain and induce a neuroprotective effect, we tested if the altered metabolites detected in the liver could also be detected in the serum of mice following mitochondrial transfer therapy. Metabolomic analysis of serum collected 13 days following the second IV mitochondrial transfer in the 6-month-old AD mice (on C57BL/6JRCCHSD background), untreated AD mice, and non-AD mice was performed. 

As presented in the heatmap (ANOVA, *p* < 0.05) (Figure 10A), C24:1-OH sphingomyelin displayed a Treatment Rescued effect and Asp showed a Treatment Effect in the serum. These metabolites showed the same effect in the liver. The serum C24:1-OH sphingomyelin and Asp response detected in the heatmap is also presented in the direct comparison of the levels of the significantly altered metabolites in the untreated AD vs. treated AD mice (Appendix A). A trend toward difference in the levels of carnosine and putrescine was also noted. Further confirmation came from the VIP scores (Figure 10B) and correlation analysis of the untreated AD and treated AD mice (Figure 10C), which also pointed to the alterations in Ala, Asp, Asn, Gly, Glu, and hexose. These results indicate that among the fifteen altered metabolites evident in the liver following the mitochondrial treatment, eight were also altered in the serum of treated mice (Table 2) (comparisons based on alterations in the heatmap, metabolite level comparison, top VIP score, and correlating metabolites). 

## 4. Discussion

### 4.1. Beneficial Effects of the Mitochondrial Transfer Therapy in the 5XFAD Mice

As a proof of concept for AD, we recently showed for the first time the beneficial effect of one IV injection of fresh and active mitochondria in the short-term pharmacological AD mouse model (ICV-injected Aβ) [33]. We now report the efficacy of the mitochondrial transfer therapy in the long-term chronic model for AD, the 5XFAD Tg mouse with amyloid pathology. We provide mechanistic insight into the effect of IV-injected exogenous mitochondria on the diseased brain without their internalization into the brain.

The IV transfer of mitochondria (repeated injections) to the 5XFAD mice ameliorated cognitive impairments and reduced neuronal damage and amyloid burden, with improvement in the mitochondrial enzymatic activity. 

Proteomic studies of the hippocampus pointed to alterations following treatment in proteins, which may lead to the amelioration of cognitive impairment, amyloid burden, and mitochondrial dysfunction. Starting with the proteomic, alterations associated with processes and components involved in cognition are (a) synaptic-related processes and GABA-regulation (with altered proteins such as extrasynaptic subunits GABA receptor subunit alpha (Gabra 3–5), part of the GABAergic system which is involved in cognitive impairment in AD) [51,52], (b) alteration in Bsn (a presynaptic protein fundamental in neurotransmitter release) [53], (c) the post-synaptic density (PSD) scaffolding proteins Dlg4 and Shank1 (members of PSD proteins, which are essential for the proper function of both the ionotropic and the metabotropic glutamate receptors, regulating synaptic plasticity) [54], (d) alterations in the HDAC complex (the enzyme that participates in lasting synaptic plasticity) [55]. These alterations in the proteome may lead to the lowered cognitive impairment detected in the mitochondria-treated 5XFAD mice.

Proteomic alterations in processes and components associated with reduced amyloid burden (Aβ 1–40 and 1–42 peptides) point mostly to degradation of the amyloid depositions by: (a) the proteasome, as suggested by alteration in proteasome-related proteins (such as Psma3) and protein ubiquitination processes (presented in proteins such as Mib2, Praja-1, Rnf34), (b) phagocytosis, as suggested by proteomic alteration of positive regulation of phagocytosis, with the regulation of Fc receptor signaling, (c) alterations in the HDAC complex, which may affect amyloid degradation, as suggested by the report that the modulation of HDAC activity affected the expression of the Aβ-degrading metalloprotease, neprilysin [56]. In addition, the alteration in phosphorylation of Gsk3a (reducing its activity), the enzyme which is involved in the regulation of Aβ production [57] and was detected in the hippocampus of the mice treated with mitochondria, may suggest that some decrease in the production of Aβ is also taking place. 

Proteomic alterations associated with the increase in mitochondrial enzymatic activity COX/CS in the brain (increase in oxidative phosphorylation) are the alterations in electron transfer oxidoreductase Ntfdh, Ndufa11 (an accessory subunit of Complex I), and Cox6a1, as well as Coa3 and Surfs (both participating in COX assembly). Regulation of mitochondrial fission (a process needed to create new mitochondria and for quality control by enabling the removal of damaged mitochondria) was also noted in the alteration of the proteome, such as Marchf5 (a ubiquitin ligase of the mitochondrial outer membrane) and Mief1, as well as mitochondrial translation by Myg1, Coa3, and Mief1. Additional mitochondria-altered proteins are Shmt2, Mrpl38, Mrps6, the phosphorylation PTM-altered Pdha1 (which catalyzes the conversion of pyruvate to acetyl-CoA and thereby links the glycolytic pathway to the tricarboxylic cycle), and Fxn (involved in the assembly of iron-sulfur clusters) [58]. These alterations in the proteome may alleviate the mitochondrial dysfunction detected in the mitochondria-treated 5XFAD mice.

Alterations in mRNA splicing and transcription regulation in the brain may point to the effect of the mitochondrial transfer treatment at the RNA level, for example, the impact on mRNA splicing via the spliceosome and the negative regulation of mRNA metabolic processes (presented in proteins such as Med1, Med20, and Bclaf1). This, and the alteration in the protein translational initiation process (shown in proteins such as Eif4a2, Eif4h, Eif3h, and Abce1), may indicate attempts to regulate the synthesis of relevant proteins following the mitochondria treatment. 

The effect of mitochondrial transfer therapy in the brain on blood coagulation, hemostasis, and erythrocyte development is also interesting and can be noted by the alteration in proteins such as Fgb, Pros1 (an anticoagulant plasma protein), and Slc4a1 (a major integral membrane glycoprotein of the erythrocyte membrane that is required for normal flexibility and stability of the erythrocyte membrane). The involvement of these processes in the response to treatment is relevant to the vascular pathology involved in AD. The pathologies include an influx of plasma proteins into the CNS through the damaged BBB, with fibrinogen in the CNS inducing immune reactions (such as microglial activation) that lead to neurodegeneration and also acting as an active contributor to neurodegenerative diseases such as AD, with altered hemostasis (with Aβ-fibrin(ogen) binding) that could thus contribute to Aβ deposition [59]. 

Exogenously injected mitochondria were detected in the liver but not in the brain. This is in accord with our results after IV mitochondrial transfer in the pharmacological ICV-injected Aβ model of AD [33]. As the organ that receives the highest blood supply, the accumulation of the IV-injected mitochondria in the liver is reasonable. The fact that the IV-injected mitochondria were not evident in the brain suggests that the beneficial effect of the mitochondrial transfer is not mediated by the direct entry of the mitochondria into the brain. These results are supported by other in vivo reports showing a limited exogenous mitochondria internalization into the target cells in cardiomyocytes and the retina [19,21,30,31,32]. This phenomenon cannot explain the beneficial effect of mitochondria therapy via mitochondria uptake by the target organ, making the mechanism of this effect very intriguing. 

Studying the response to the mitochondrial transfer by the liver, the organ in which the IV-injected mitochondria were detected, revealed alteration in the proteins Pik3r1, Gstp3, Igfbp2, Glb1, and others, with alterations in the metabolites Asn, Asp, Ala, Gly, Glu, carnosine, putrescine, hexose, C24:1-OH sphingomyelin, hexanoylcarnitine, and others.

Of high interest are the alterations in insulin-like growth factor-binding protein-2 (Igfbp2), an abundant Igf-binding protein among the group of high-affinity Igfbps, which are the main regulators of the bioavailability and functions of Igf-l/Igf-ll in the circulation. Igfbp2 binds to Igfs and interferes with their binding to Igf-IR to control their bioavailability and localization. As a result, they inhibit the action of the Igf, a growth factor with neuromodulatory activities in the CNS, including the promotion of synapse formation, regulation of metabolic functions such as glucose uptake in glial cells, and involvement in memory enhancement and consolidation, via PI3K signaling pathways [60]. As a serum neurotrophic factor exerting a tonic trophic input on brain cells, providing a mechanism for what may be referred to as neuroprotective surveillance [61], it may be hypothesized that decreased Igfbp2 may lead to an increased systemic bioavailability of the Igf, thereby allowing its arrival to the brain to induce protective effects. A protein that was highly elevated in response to treatment was R3hcc1, which was predicted to enable nucleic acid binding activity that may be related to changes in gene expression. These proteome alterations point to a broad metabolic reaction of the liver. 

Integrating the changes discovered in the metabolomic and the proteomic profiles of the liver in response to mitochondrial transfer therapy revealed alterations in amino acid metabolism, glutathione metabolism, glucose metabolism, sphingolipid metabolism, and acylcarnitines (detailed in Table 1). The accordance between liver metabolomic and proteomic alterations in response to the mitochondrial transfer therapy further supports the broad metabolic reaction of the liver.

When testing if the altered metabolites detected in the liver are also detected in the serum of mice following mitochondrial transfer therapy, we discovered that among the fifteen altered metabolites evident in the liver following the mitochondrial treatment, eight were also altered in the serum of treated mice: Asp, Ala, Gly, Glu, carnosine, putrescine, hexose, and C24:1-OH sphingomyelin.

### 4.2. A Possible Liver–Serum–Brain Route in the Mitochondrial Transfer Mechanism

The finding that metabolites are altered in the liver and also in the serum may suggest that the beneficial changes in the brain are derived from the liver response to the treatment via a liver–blood–brain route. The liver/serum-altered metabolites being associated with beneficial/positive effects in brain may support such a scenario. Of high interest among these liver/serum-altered metabolites is the C24:1-OH sphingomyelin (decreased in liver and serum), which belongs to the sphingolipids, a class of lipids detected in most tissues, with higher concentrations found in nerve tissues and red blood cells. Sphingolipids are implicated in the processing and aggregation of Aβ in membrane rafts and in the transmission of the Aβ cytotoxic signal in AD [62]. Sphingolipids are also proposed prognostic biomarkers of neurodegeneration [63]. Other relevant metabolites are the amino acids Ala, Asp, Asn, Gly, and Glu (increased in liver and serum). These metabolites serve as neurotransmitters or their precursors. The altered biogenic amines of putrescine (increased in the liver and serum) and carnosine (decreased in liver and increased in serum) are also interesting, as they have been reported to have neuroprotective effects. Carnosine is involved in anti-aggregation, antioxidant, and anti-inflammatory activities, the prevention of amyloid-related neurodegeneration [64], the reduction of amyloid burden in AD mice, and the improvement of cognition and mitochondrial dysfunction in AD mice [65] and age-related dementia mice [66]. Low levels of putrescine are associated with declining memory abilities in fruit flies and rats [67] and are decreased in the temporal cortex of AD patients with activation of NMDA receptor [68]. Alteration in the monosaccharide hexose (90% glucose) is also relevant, as it plays a central role in energy homeostasis. The fact that carnosine and hexose manifested an inversed response to mitochondria injection in AD mice (carnosine decreased in liver and increased in serum, and hexose increased in liver and decreased in serum) may suggest the involvement of active transporters, mediating their transfer against the concentration gradient.

Transfer of mitochondria is a naturally occurring physiological process (“mitochondrial exchange”). Neurons can release damaged mitochondria and transfer them to astrocytes for disposal and recycling [69]. The ability to exchange mitochondria may represent a potential mode of cell–cell signaling in the CNS, and astrocytes could provide a source for functional mitochondria that enter into neurons. This exchange between neurons and astrocytes occurs via tunneling nanotubes, extracellular vesicles, or cell fusion [70]. However, the precise mechanism by which the exogenously injected mitochondria induce a beneficial effect is still unclear. In particular, the contribution, or necessity, of the internalization of the injected exogenous mitochondria into a germane tissue target is not fully understood. 

In cell culture, exogenous mitochondrial internalization was shown to be mediated by macropinocytosis or endocytosis [16,17,18]. However, in vivo, e.g., in cardiomyocytes of ischemia-reperfusion injury, the mechanism of mitochondrial internalization is not fully understood. An improvement in cardiac functions was observed 10 min after mitochondrial delivery (excluding a DNA transfer effect), with the number of exogenous mitochondria detected in cardiomyocytes being very low and significantly less than in vitro (only 3–7% were internalized). It is hard to understand how a few mitochondria can substantially influence energy supply in cardiomyocytes [19,21,30,32,71,72] and have a beneficial effect in optic nerve injury, while only a few neurons internalize exogenous mitochondria in the RGC layer of the retina [31]. The apparent discrepancy between the very low level of mitochondria internalization into target tissues and the significant biological protective effects observed raises important questions regarding the mechanism of mitochondrial transfer therapy in these models. In the current study of IV mitochondrial transfer in 5XFAD mice, and in our previous study in the AD ICV-Aβ model [33], improvement of AD symptoms and brain pathology was evident as well, yet the presence of the injected mitochondria in the target organ—the brain—was not detected. However, while the mechanism of mitochondria therapy in heart ischemia/reperfusion or other models is still not fully understood, we present here the mechanistic aspects of mitochondrial transfer therapy in AD Tg mice. Our current finding of altered metabolites in liver and serum of AD mice treated with IV mitochondrial transfer therapy, and the fact that these metabolites are neuroprotective, might suggest a scenario of liver response to the IV injection followed by regulated secretion of metabolites into the circulation, which then arrive to the brain via a liver–blood–brain axis. These metabolites may directly or indirectly affect disease pathogenesis, thereby leading to a beneficial anti-AD effect, manifested as changes in cognition, amyloid burden, mitochondrial function, and RNA regulation. This scenario agrees with the established phenomenon of cross-talk between the liver and the brain, such as via metabolic/endocrine signals in hepatic encephalopathy, a brain dysfunction caused by liver insufficiency, and in the nonalcoholic fatty liver disease associated with cognitive changes and brain volume reduction [73]. Moreover, the involvement of growth factors in the cross-talk between the liver and brain has been reported, particularly the fibroblast growth factors mediating the liver–brain cross talk during prolonged fasting [74]. This agrees with our results showing an increase in the liver in Igfbp2, a regulator of the bioavailability of the circulating neuroprotective growth factor Igf in the treated AD mice. Our previous results in the ICV-Aβ AD model [33], showing that the response of the mitochondrial enzymes (increased activity) in the liver precedes this mitochondrial response in the brain, are in accord with our suggested scenario of a liver–blood–brain-mediated response of the IV mitochondrial transfer therapy.

Taken together, our proposed mechanism for the beneficial anti-AD effect of IV mitochondrial transfer is that the mitochondria injected into the tail vein arrive (via the heart) to the liver (the organ with the highest blood supply). In the liver, they induce a wide metabolic response, leading to a regulated secretion of metabolites into the circulation, which then arrive to the brain via a liver–blood–brain metabolic axis. These metabolites, which are known to have neuroprotective properties, directly or indirectly affect disease pathogenesis, thereby leading to a beneficial anti-AD effect, manifested as changes in cognition, amyloid burden, mitochondrial function, and RNA regulation.

### 4.3. Mitochondrial Transfer—Clinical Implications

Interestingly, the beneficial effect was noted in mice that started therapy when they were 6 and even 12 months old, with two to four IV injections of mitochondria. This finding points to the broad efficacy of this therapeutic approach and has a promising potential for use at early and even later stages of the disease. It will be important to determine how long the beneficial effect of an optimal protocol lasts, including frequency and number of injections. 

Although we did not encounter any safety problems [33], when further developing this therapeutic approach toward clinical trials, safety aspects need to be further addressed. While treating individuals under acute injury condition by mitochondria therapy demands accurate timing in being as close as possible to the injury time to achieve effective results, applying this therapeutic approach in the clinic for a chronic neurodegenerative disease like AD may be feasible and worthwhile. 

## 5. Conclusions

Our results, showing the high efficacy at ameliorating the disease in the genetic AD mice, strongly support our new concept of treating AD with mitochondria IV transfer. Based on proteomic and metabolomic alterations, we provide a mechanistic vision of the enigmatic phenomena of a beneficial effect of mitochondrial transfer not necessarily mediated by internalization into the target cells, suggesting a liver-mediated response via a metabolic liver–blood–brain axis. Mitochondrial transfer can be applied to humans, since collecting mitochondria from various cell types, including autologous sources, is a feasible procedure. 

## Figures and Tables

**Figure 1 cells-12-01006-f001:**
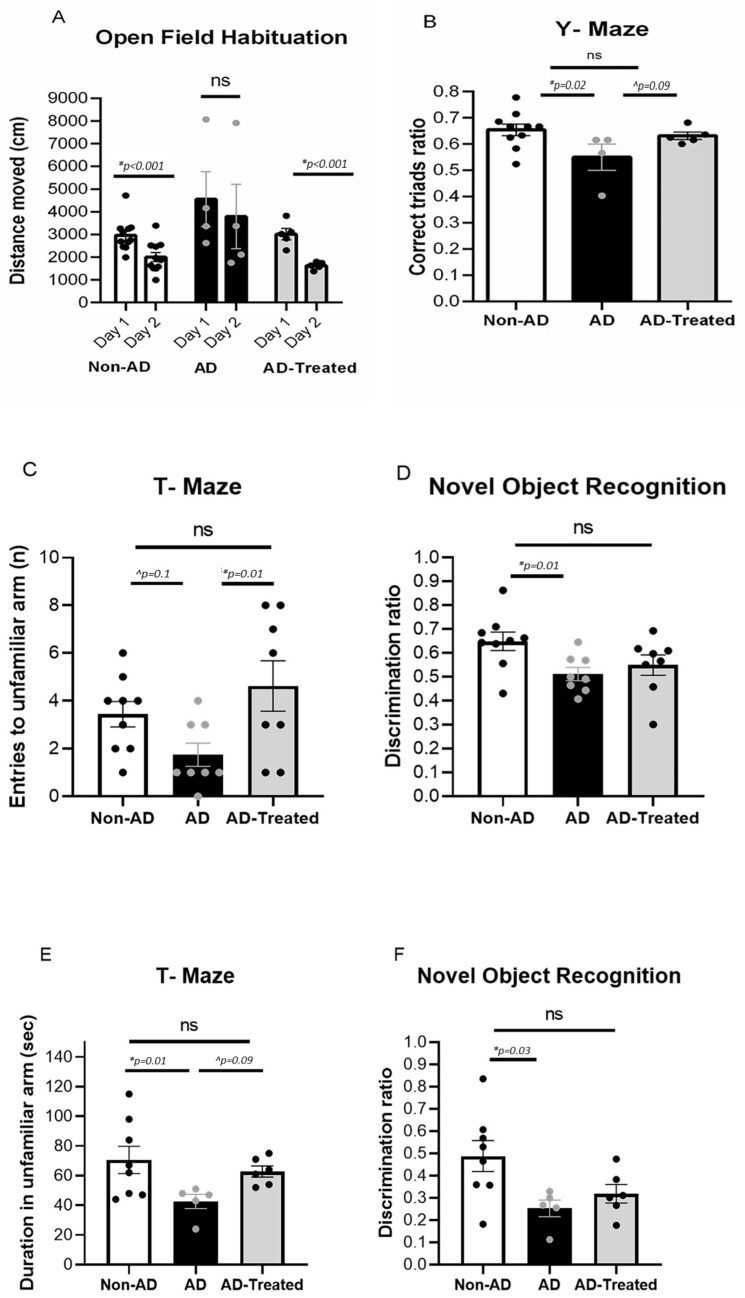
**IV mitochondria-treated AD mice performed better in cognitive tests than untreated AD mice.** (**A**,**B**) AD mice that received four injections of mitochondria starting at 6 months of age showed better cognitive performance compared to untreated mice, approaching the performance of non-AD mice. In the Open Field Habituation test (**A**), there was a significantly lower distance walk in Day 2 compared to Day 1 in the non-AD mice (*t*-test, *p* < 0.001) and in the treated AD mice (*p* < 0.001), but not in the untreated AD mice. In the Y-maze test (**B**), one-way ANOVA showed a trend toward difference between the groups [f(2, 16) = 3.22, *p* = 0.06]. LSD post-hoc analysis showed significantly worse performance (lower correct triads ratio) in AD mice relative to the non-AD group (*p* = 0.02), and a trend toward better performance in treated AD mice relative to untreated AD mice (*p* = 0.09), with no difference between treated AD mice and non-AD mice. (**C**,**D**) AD mice that received three injections of mitochondria starting at 6 months of age showed better cognition compared to untreated AD mice, similar to the performance of non-AD mice. In the T-maze test (**C**), one-way ANOVA revealed a significant difference between the groups [f(2, 22) = 3.84, *p* = 0.037]. LSD post-hoc analysis showed significantly better performance (higher number of entries into the new arm) in treated AD mice relative to untreated AD mice (*p* = 0.01) and a trend toward worse performance in AD mice relative to the non-AD group (*p* = 0.1), with no difference between treated AD mice and non-AD mice. There was better cognitive performance in the mitochondria-treated AD mice in the Novel Object Recognition test (**D**) compared to untreated AD mice, similar to the performance of non-AD mice: one-way ANOVA revealed a significant difference between the groups [f(2, 22) = 3.76, *p* = 0.039]. LSD post-hoc analysis showed significantly worse performance (lower discrimination ratio: spending less time near the novel object) in AD mice relative to the non-AD group (*p* = 0.01), with no difference between treated AD mice and non-AD mice. (**E**,**F**) AD mice that received two injections of the isolated mitochondria starting at 12 months of age showed better cognitive performance compared to untreated AD mice in the T-maze test. (**E**) One-way ANOVA showed a significant difference between the groups [f(2, 16) = 3.477, *p* = 0.05]. LSD post-hoc analysis showed significantly worse performance in AD mice relative to the non-AD group (*p* = 0.01) and a trend toward worse performance of AD mice relative to the treated AD group (*p* = 0.09), with no difference between treated AD mice and non-AD mice. There was better cognitive performance in the Novel Object Recognition test compared to untreated AD mice, similar to the performance of non-AD mice. (**F**) One-way ANOVA revealed a significant difference between the groups [f(2, 16) = 4.463, *p* = 0.029]. Tukey post-hoc analysis showed significantly worse performance in AD mice relative to the non-AD group (*p* = 0.03), with no difference between treated AD mice and non-AD mice (* statistically significant; ^ trend, ns= non significant).

**Figure 2 cells-12-01006-f002:**
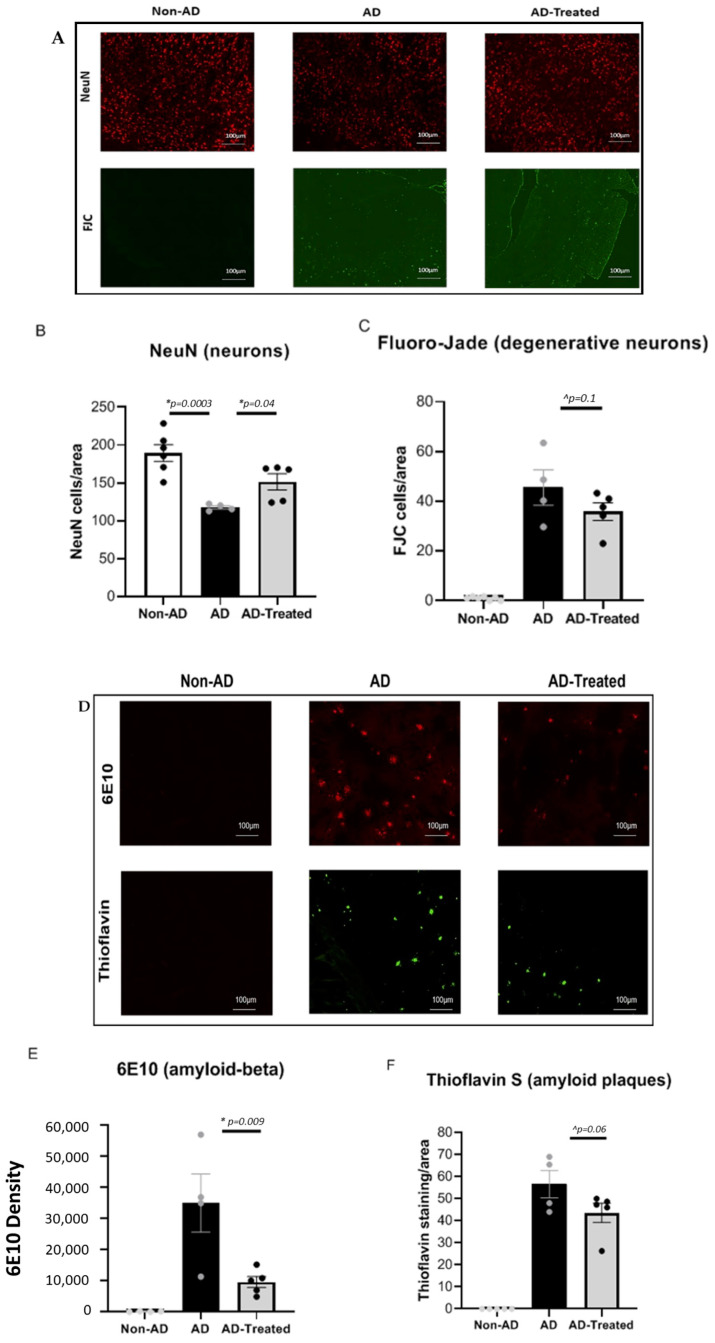
**Reduced neuronal damage and amyloid burden in the cortex of AD mice treated with mitochondrial transfer.** (**A**) NeuN and FJC staining of neurons. Lower neuronal count (NeuN) (relative to non-AD mice) and presence of degenerative neurons (FJC) were detected in the AD mice, with increased neural counts and decreased degenerative neurons in mitochondria-treated AD mice relative to untreated AD mice. (**B**,**C**) Quantitative analysis: one-way ANOVA showed a significant difference between the groups [f(2, 12) = 12.59, *p* = 0.001]. LSD post-hoc analysis showed a significantly lower neuronal count in AD vs. non-AD mice (*p* = 0.0003), with higher count in mitochondria-treated vs. untreated AD mice (*p* = 0.04) and a trend toward decrease in degenerative neurons in the treated vs. untreated AD mice (*t*-test, *p* = 0.1). (**D**) 6E10 and thioflavin staining for amyloid. Amyloid pathology was detected in AD mice. Decreased amyloid pathology in the mitochondria-treated relative to untreated AD mice. (**E**,**F**) Quantitative analysis: lower amyloid burden (6E10) and plaques (thioflavin) in mitochondria-treated vs. untreated AD mice (*t*-test, *p* = 0.009; trend, *p* = 0.06, respectively). Staining of 6E10 and thioflavin was hardly detected in the non-AD mice (* statistically significant; ^ trend).

**Figure 3 cells-12-01006-f003:**
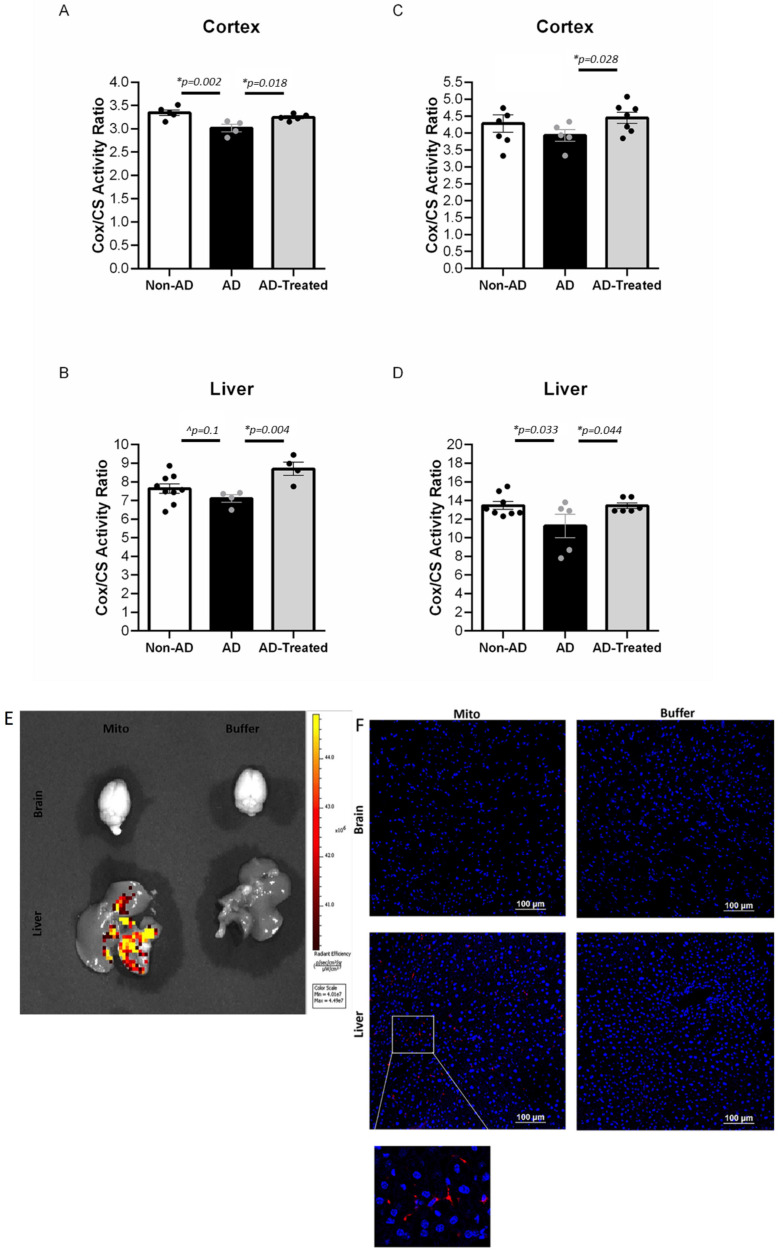
**Increased mitochondrial enzymatic activity in the brain and liver of AD mice treated with mitochondrial transfer (A–D). Injected mitochondria are detected in the liver and not in brain (E,F).** (**A**,**B**) COX/CS activity ratio in the brain and liver of 6-month-old treated mice (four injections). (**A**) One-way ANOVA showed a significant difference in brain between the groups [f(2, 11) = 8.064, *p* = 0.007]. LSD post-hoc analysis showed a significantly lower ratio in AD vs. non-AD mice (*p* = 0.002), with higher ratio in mitochondria-treated vs. untreated AD mice (*p* = 0.018), reaching the ratio of the non-AD mice. (**B**) A higher COX/CS ratio was detected in the liver of treated AD mice relative to the untreated AD mice (*t*-test, *p* = 0.004), reaching even a higher value than in the liver of the non-AD mice, with a trend showing a reduced ratio in the AD relative to non-AD mice (*p* = 0.1). (**C**,**D**) COX/CS activity ratio in the brain and liver of 12-month-old treated mice (two injections). (**C**) The treated AD mice showed a higher COX/CS ratio in brain relative to the untreated AD mice (*t*-test, *p* = 0.028), reaching the ratio of the non-AD and even exceeding it, with a trend showing a reduced ratio in the AD relative to non-AD mice (*p* = 0.15). (**D**) One-way ANOVA showed a trend toward difference in brain between the groups [f(2, 16) = 3.245, *p* = 0.06]. LSD post-hoc analysis showed a significantly higher COX/CS ratio in the treated compared to the untreated AD mice (*p* = 0.044), with a reduced ratio in the AD relative to non-AD mice (*p* = 0.033). (**E**,**F**) DsRed signal is detected in the liver and not in the brain of AD mice about 2 h following the injection of DsRed mitochondria. (**E**) Using IVIS: DsRed signal in the liver of mitochondria-injected (200 mgr/mouse buffer) AD mice but not in their brain (hippocampus region is presented). No signal in the brains and livers of buffer-injected mice. (**F**) Using anti-RFP Ab fluorescence staining: DsRed signal in the liver of mitochondria-injected (500 mgr/mouse) AD mice but not in their brain. No signal in the brains and livers of buffer-injected mice (* statistically significant; ^ trend).

**Figure 4 cells-12-01006-f004:**
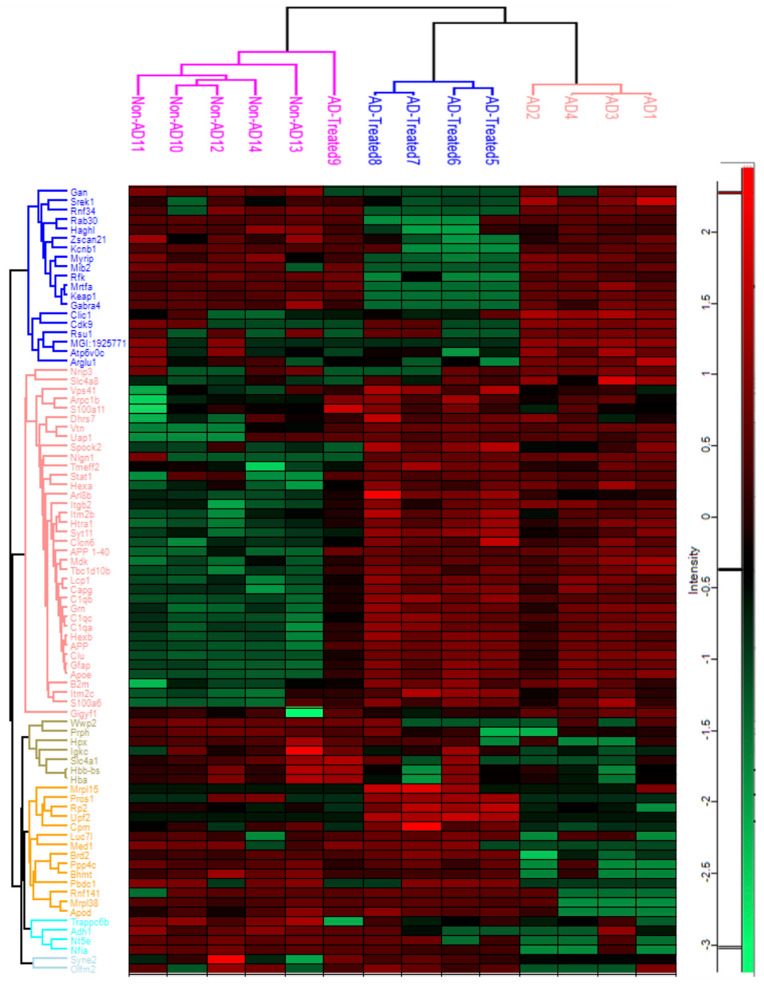
**Proteomics analysis of the hippocampus: heatmap analysis of hippocampus homogenates**. Analysis showing proteins significantly altered across all group comparisons (non-AD, AD, and treated AD). Red is indicative of upregulation while green is indicative of downregulation.

**Figure 5 cells-12-01006-f005:**
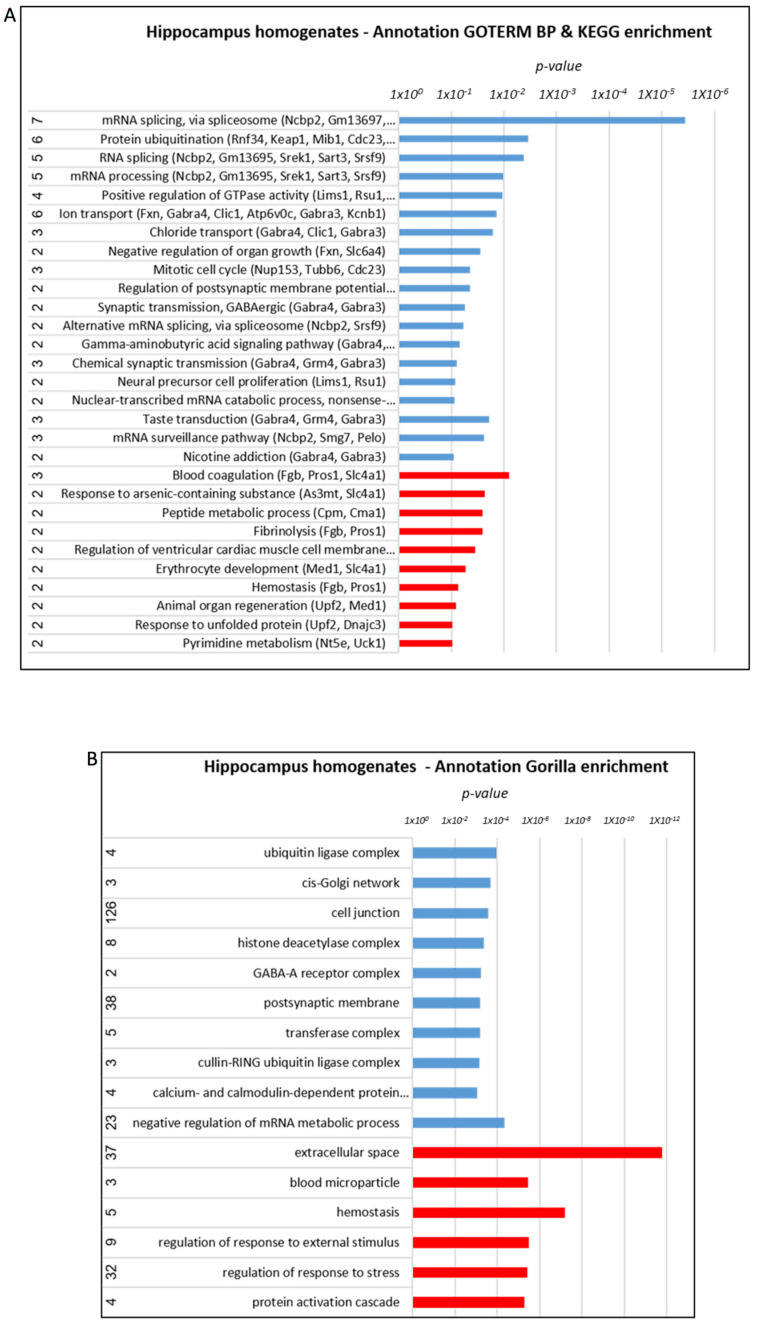
**Proteomics analysis of the hippocampus: annotation analysis of hippocampus homogenates.** (**A**) KEGG/GOTERM (names of affected proteins presented in brackets; full names of proteins are presented in Appendix A). (**B**) Gorilla enrichment. AD > treated AD—marked in blue, AD < treated AD—marked in red.

**Figure 6 cells-12-01006-f006:**
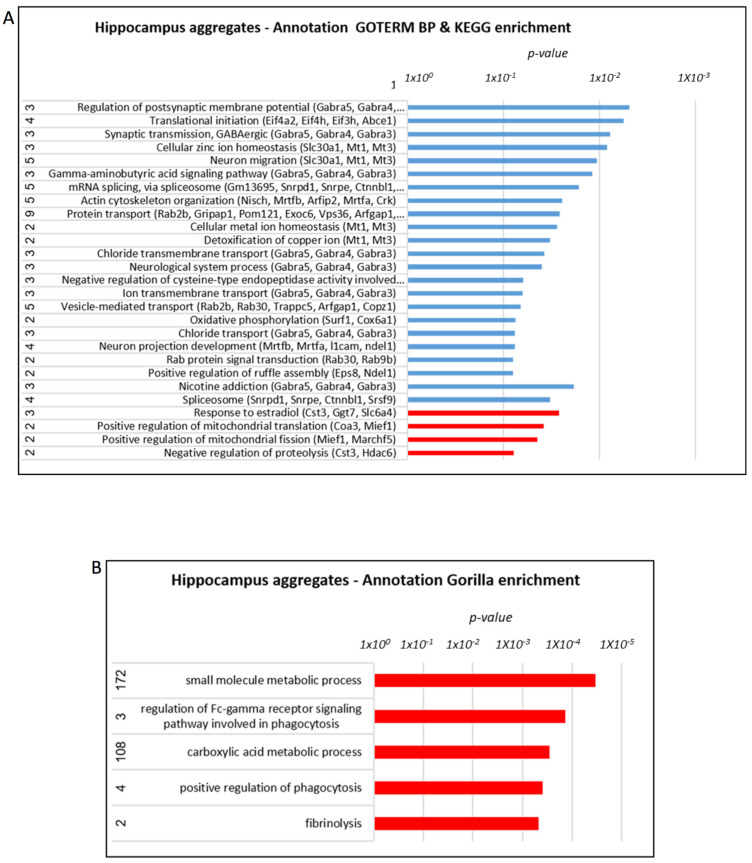
**Proteomics analysis of the hippocampus: annotation analysis of hippocampus aggregates.** (**A**) KEGG/GOTERM (names of affected proteins presented in brackets; full names of proteins are presented in Appendix A). (**B**) Gorilla enrichment AD > treated AD—marked in blue, AD < treated AD—marked in red.

**Figure 7 cells-12-01006-f007:**
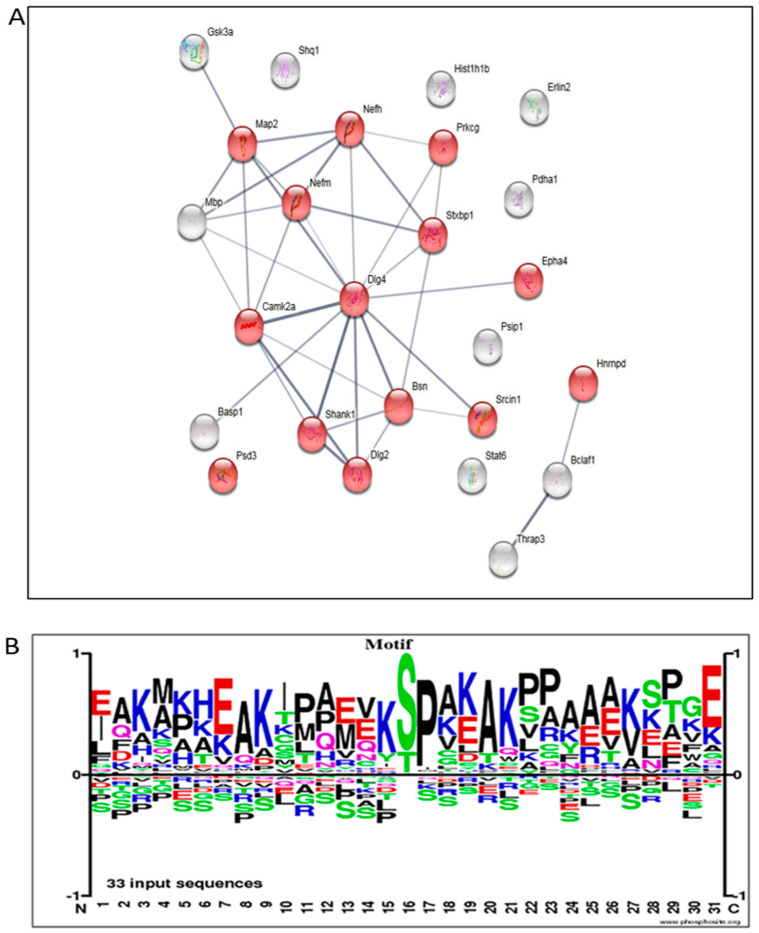
**Proteomics analysis of the hippocampus: PTM-phosphorylation analysis.** (**A**) STRING analysis of close clustering of synapse related proteins (in red) (full names of proteins are presented in the Appendix A). (**B**) Identification of the amino-acid motifs of the differential phospho-peptides between the AD and treated AD groups in the brain samples.

**Figure 8 cells-12-01006-f008:**
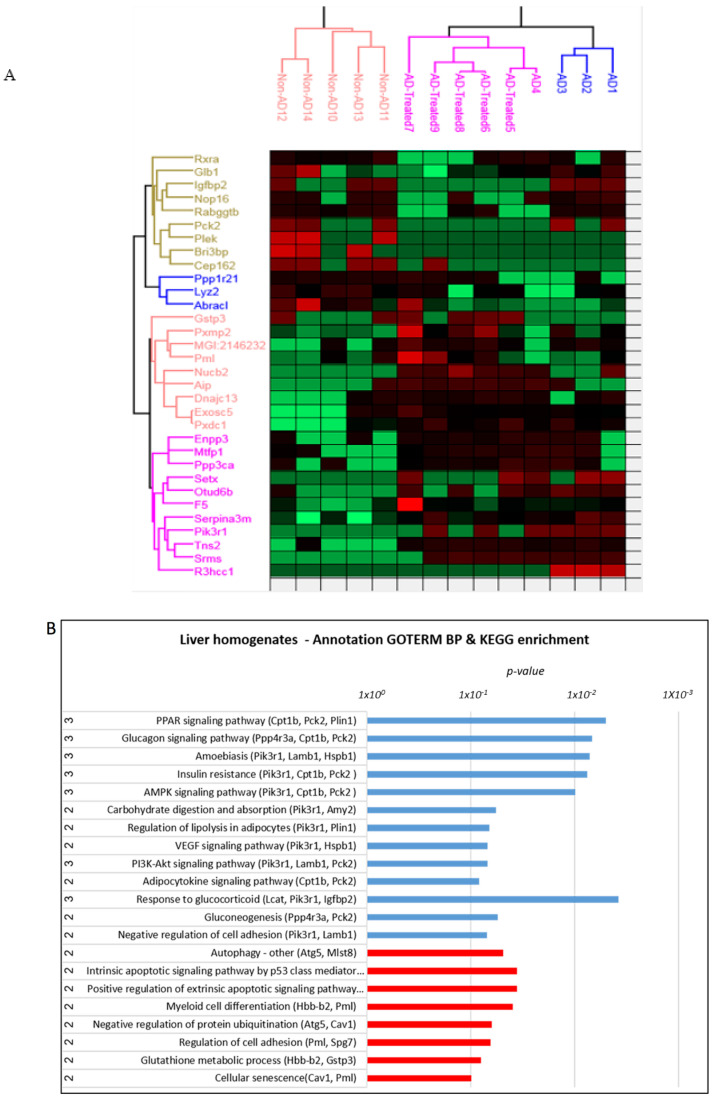
**Proteomics analysis of the liver.** (**A**) Heatmap showing proteins significantly altered across all group comparisons (non-AD, AD, and treated AD) in liver homogenates. Red is indicative of upregulation while green is indicative of downregulation. (**B**,**C**) Annotation analysis. (**B**) KEGG/GOTERM (names of affected proteins presented in brackets; full names of proteins are presented in Appendix A). (**C**) Gorilla enrichment of the liver. AD > treated AD—marked in blue, AD < treated AD—marked in red.

**Figure 9 cells-12-01006-f009:**
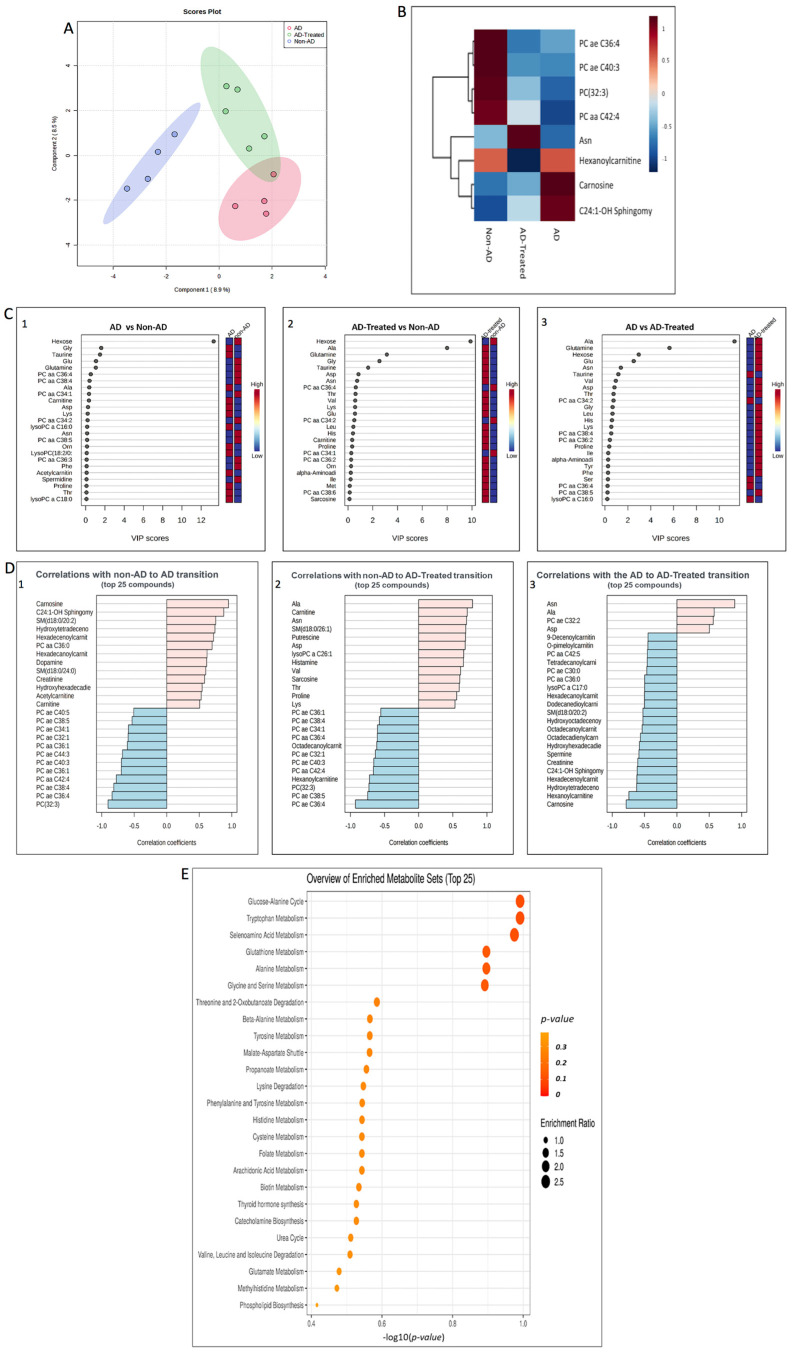
**Metabolomics analysis of the liver.** (**A**) PLS-DA (“supervised PCA”) obtained from the three groups (non-AD, AD mice, and treated AD mice) showed separation of the samples, with the AD mice and the non-AD mice showing the biggest difference, while the treated AD group separates from the AD mice. (**B**) Heatmap showing metabolites significantly altered across all group comparisons, presented as average value for each group. (**C**) VIP score plots, presenting the contribution of the metabolites (top 25) with the highest impact on the difference between the tested groups, showing Treatment Rescued effect (such as hexose, taurine, glutamine, Glu) or Treatment Effect (such as Asn, Asp, Ala). (Detailed VIP contribution in Appendix A). (**D**). Correlation of the metabolites (top 25) with the transitions (differences) between the groups (non-AD vs. AD, non-AD vs. treated AD and AD vs. treated AD). Positive correlation marked in red, negative correlation marked in blue. Most of the metabolites showing positive correlation in the non-AD vs. AD do not show such a correlation in the non-AD vs. treated-AD and also show negative correlation in the AD vs. treated AD (such as carnosine, sphingomyelin, hydroxytetradeceno, and hexadecenolcarnitine), thereby pointing to a Treatment Rescued effect. A Treatment Effect in the AD vs. treated AD difference showed correlated metabolites (such as Ala, Asn, Asp). (**E**) Metabolomic Pathway Analysis (SMPDB pathway libraries as references) for identifying the metabolic pathways enriched by the mitochondria treatment. The 5–6 top enriched sets in the treated AD vs. the AD mice are highly significant.

**Figure 10 cells-12-01006-f010:**
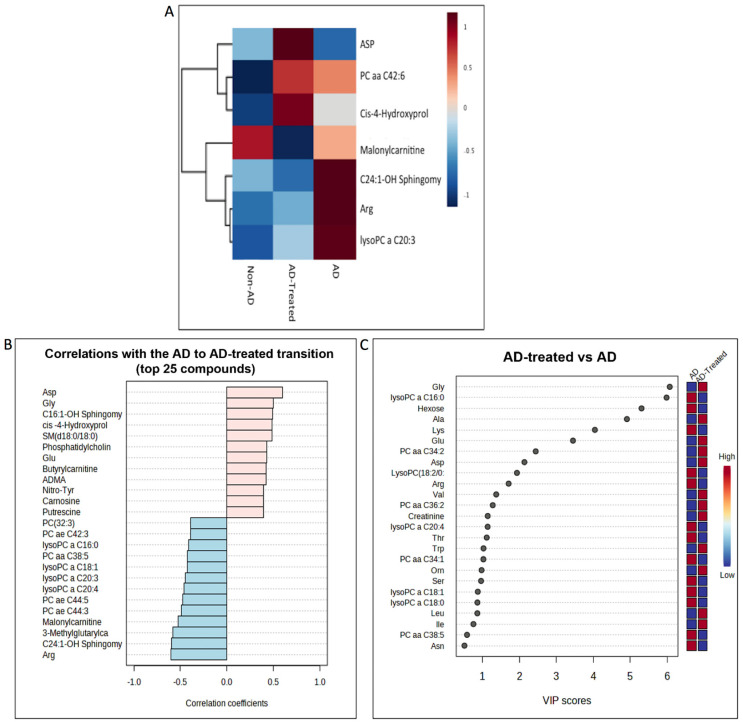
**Metabolomics analysis of the serum.** (**A**) Heatmap showing metabolites significantly altered across all group comparisons, presented as average value for each group. (**B**) VIP score plot and (**C**) correlation of metabolites with treated AD vs. the AD difference point to the alterations in Ala, Asp, Asn, Gly, Glu, and hexose.

**Table 1 cells-12-01006-t001:** Integrative analysis of proteomics and metabolomics in the liver.

	Metabolomics	Proteomics
**Amino acid metabolism**	Altered metabolites:	Enriched terms: Cellular amino acids metabolic process, small molecule metabolic/biosynthetic process
	Asn, Asp, Ala, Gly, Glu, glutamine, carnosine, putrascine.	
	Enriched pathways:	
	Tryptophan metabolism, alanine metabolism, glutathione metabolism, glycine and serine metabolism	
**Glutathione metabolism**	Enriched pathways: Glutathione metabolism	Altered proteins:
		Gstp3
		Enriched pathways/terms:
		Glutathione metabolic process
**Glucose metabolism**	Altered metabolites: Hexose Enriched pathways: Glucose alanine cycle	Altered proteins:
		Pck2, Glb1, Igfbp2.
		Enriched pathways:
		PPAR signaling, glucagon signaling, insulin resistance, AMPK signaling, PI3K/AKT signaling, gluconeogenesis
**Sphingolipid metabolism**	Altered metabolites: C24:1-OH sphingomyelin SM(d18:0/20:2)	Altered proteins:
		Pik3r1, Glb1
		Enriched terms:
		Lipid biosynthetic pathway
**Acylcarnitines (and mitochondria)**	Altered metabolites: Hydroxytetradecenoylcarnitine, hexadecenoylcarnitine, hexanoylcarnitine	Altered proteins:
		Pck2
		Enriched terms:
		Mitochondrial respiratory chain complex assembly

**Table 2 cells-12-01006-t002:** Comparison of altered proteins due to mitochondrial treatment in the liver and in the serum (“increase” or “decrease” in the treated AD vs. the AD mice).

Metabolite	Liver	Serum
Acylcarnitines		
Hydroxytetradecenoylcarnitine	decrease	-
Hexadecenoylcarnitine	decrease	-
Hexanoylcarnitine	decrease	-
Sphingolipids		
C24:1-OH Sphingomyelin	decrease	decrease
SM(d18:0/20:2)	decrease	-
Amino acids		
Ala	increase	increase
Asp	increase	increase
Asn	increase	-
Gly	increase	increase
Glu	increase	increase
Glutamine	increase	-
Biogenic amines		
Carnosine	decrease	increase
Putrescine	increase	increase
Taurine	decrease	-
Monosaccharides		
Hexose	increase	decrease

## Data Availability

Not applicable.

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
