# Peer review of "The Beneficial Effect of Mitochondrial Transfer Therapy in 5XFAD Mice via Liver–Serum–Brain Response"

_cells, 2023, doi:10.3390/cells12071006_

Round 1

Reviewer 1 Report

The Sweetat et al. article addresses a very interesting topic on mitochondrial transfer therapy, but I have major and minor concerns that need to be addressed in order to consider this article for publication.

Major concerns:

1) The authors frequently mention that some results have a "trend", which is indicated in the graphs with the corresponding p-value, as in the case of significant differences. All these results indicated with a "trend" are not significant, like the other non-significant results, there is no reason to distinguish them from the other non-significant results. The use of the "trend" is biased and can lead to misinterpretation. My recommendation is to correct this point in the graphs and along the text.  A new version with these changes could be evaluated more objectively.

2) The images in figure 2 have poor quality and presentation, please address this issue. Check the quality of all figures according to the criteria of the guidelines for authors. With the current version of the images, it is not possible to evaluate the results. The description of the histological analysis methodology needs to be improved: region analyzed, how many sections were used for counting, etc.

3) Figure 4: I suggest separating this figure into several independent figures to give enough space to each one and improve its presentation, some images appear distorted. Figure 4A (heat map) needs more size, the protein names are not readable.

4) Table 2: It is not entirely clear in which condition, AD or treated AD, the increase or decrease occurs.

5) Discussion: I suggest addressing in greater depth the different mechanisms of mitochondrial transfer (intercellular and extracellular), as well as the risks and benefits of this novel technique.

Minor concerns:

1.       Correspondence is indicated with different symbols # and *

2.       The authors have to review and correct several typos throughout the text: e.g. benificial (line571), benfit (line20),…

3.       Line 23: 5XFAD “mice”?

4.       Please avoid abbreviations as much as possible in the abstract so that when it is read it can be understood without having to refer to the full text; or describe the abbreviations in the abstract itself.

5.       Table 2 appears in the discussion section instead of the results section.

1.       Line 609: …may suggest that some decrease in the production of “A?” is also taking place…( a beta symbol is missing)

Author Response

Reply to Reviewer 1:

We thank very much the reviewer for his review, providing us comments and suggestions to improve the manuscript, and allowing us to revise our manuscript. 

Please see point to point our answers to the comments:

Reviewer 1

The Sweetat et al. article addresses a very interesting topic on mitochondrial transfer therapy, but I have major and minor concerns that need to be addressed in order to consider this article for publication.

Major concerns:

  • Comment: The authors frequently mention that some results have a "trend", which is indicated in the graphs with the corresponding p-value, as in the case of significant differences. All these results indicated with a "trend" are not significant, like the other non-significant results, there is no reason to distinguish them from the other non-significant results. The use of the "trend" is biased and can lead to misinterpretation. My recommendation is to correct this point in the graphs and along the text.  A new version with these changes could be evaluated more objectively.

Reply: We understand and appreciate this comment of the reviewer. We would like to explain the reason for presenting our results in a way that gives also place to results having some lower degree of significance than the cutoff of 0.05, using the 0.1 cutoff for presenting some trend of effect.

We used the term "trend" aiming to describe results with weak evidence, similar to the definition in VSNI (data science software and experimental design software for biosciences, https://vsni.co.uk/blogs/what-is-a-p-value) : "…This leads to the typical guidelines of: p < 0.001 indicating very strong evidence against H0, p < 0.01 strong evidence, p < 0.05 moderate evidence, p < 0.1 weak evidence or a trend, and p ≥ 0.1 indicating insufficient evidence…". 

Our results presenting a trend of difference, such as in the FJC staining for degenerative neurons (decrease in the mito-treated mice), came as support to the main significant finding, with p=0.04, of increase in neuron count in the mito-treated mice (using immunostaining with the NeuN neuronal nuclei Ab). Also, staining for amyloid using the amyloid Ab (6E10), we first observed highly significant (p=0.009) decrease in amyloid peptides following mitochondrial transfer. This decrease was also demonstrated by the proteomics analysis. Further support for the mitochondrial transfer-mediated reduction in the amyloid burden is provided by thioflavine staining even though it reached a level of significance of only 0.06. This observation reflects the same phenomenon indicated by the more significant 6E10 results.

Moreover, the response of degradative pathways to the treatment indicated by our proteomics results (such as autophagy and phagocytosis, which degrade amyloid) further supports the results of decreased amyloid by mitochondrial transfer therapy. Similarly, in the behavioral studies, based on results of 3 independent experiments using various cognitive tests, a significant improvement of cognition in mitochondria treated mice – was noticed. Using the cutoff 0.1 indeed provides only a weak evidence, however it further supports the significant results using 0.05 cutoff, which showed that the treatment ameliorated cognitive impairment, or that while the non-AD mice had better cognitive performance than the AD-mice, the performance of these non-AD mice did not differ from that of the treated AD-mice. Further supporting the rational of presenting also the weak evidence (trend) results is the example of the T-maze performed in 2 independent experiments. In one experiment the treated mice had significantly (p=0.01) better performance than the untreated mice (treated-AD vs AD), while in the other experiment this was only a trend with p=0.09 (Fig. 1C,E). This strongly suggests that the trend observed (p=0.09) is in line with the significant result (p=0.01) using the same therapy in a repeated experiment. Similar to the difference between treated and untreated AD mice, the difference between the non-AD and AD mice in the T-maze comparison is also significant in one experiment, and a trend in the other. Not to mention that the difference between AD and non-AD mice (worse in AD, 5XFAD mice) is a general known finding. Moreover, the proteomics studies which showed synaptic and other cognitive related responses to treatment, provide further support to the improved cognitive performance.

In summary, the existence of marginally significant (p<0.1) results corroborating strongly significant (p<0.05) results led us to present in addition to the significant results also the results of weaker significance (trend), an approach which is in common use by other research groups. Some representative examples:

Eliminating microglia in Alzheimer's mice prevents neuronal loss without modulating amyloid-β pathology. Spangenberg EE, Lee RJ, Najafi AR, Rice RA, Elmore MR, Blurton-Jones M, West BL, Green KN. Brain. 2016 Apr;139(Pt 4):1265-81. doi: 10.1093/brain/aww016. (This study used the same 5XFAD mice we did). 

Tau passive immunization inhibits not only tau but also Aβ pathology. Dai CL, Tung YC, Liu F, Gong CX, Iqbal K. Alzheimers Res Ther. 2017 Jan 10;9(1):1. doi: 10.1186/s13195-016-0227-5.

Alleviation of a polyglucosan storage disorder by enhancement of autophagic glycogen catabolism. Kakhlon O, Vaknin H, Mishra K, D'Souza J, Marisat M, Sprecher U, Wald-Altman S, Dukhovny A, Raviv Y, Da'adoosh B, Engel H, Benhamron S, Nitzan K, Sweetat S, Permyakova A, Mordechai A, Akman HO, Rosenmann H, Lossos A, Tam J, Minassian BA, Weil M. EMBO Mol Med. 2021 Oct 7;13(10):e14554. doi: 10.15252/emmm.202114554]. (Collaboration with our group).

Incorporating the reviewer criticism, we defined in the Methods Statistics section that 0.05 cutoff represents a significant difference, and 0.1 cutoff represents a difference of weak significance (trend) and added a relevant reference. We added the following text: "Statistical significance was accepted at p<0.05 (*) and trends at p<0.10 (^) [Spangenberg, Brain 2016]". 

Accordingly, we removed p=0.15 from fig.3C.

2) Comment: The images in figure 2 have poor quality and presentation, please address this issue. Check the quality of all figures according to the criteria of the guidelines for authors. With the current version of the images, it is not possible to evaluate the results. The description of the histological analysis methodology needs to be improved: region analyzed, how many sections were used for counting, etc.

Reply: We now improved the quality of the photos according to the criteria of the guidelines.

We now added more details in the description of the histological studies, including details from collecting of tissues and the histological analysis, as follows: "After finalizing the behavioral tests (about 2 weeks after last mitochondrial injection) mice were anesthetized with a lethal dose of pentobarbital and perfused via the ascending aorta with ice-cold PBS. Brains and livers were removed. One-half of the brain was fixed in 4% paraformaldehyde for histological studies, and the other half was stored at −80 °C for biochemical and proteomics studies. Similarly, part of liver was fixed for histological studies, and another part was stored at −80 °C for biochemical and proteomics / metabolomics studies.

Brain sections were stained for amyloid burden (6E10, Thioflavin S) and neuronal loss (anti-neuronal nuclei, NeuN, Fluoro-Jade). Brain and liver sections were stained for the presence of the DsRed-2-labelled mitochondria (anti-red fluorescent protein, RFP). Analysis was performed in a blinded manner, analyzing three sections per animal. For 6E10 analysis, signal intensity was integrated to measure fluorescence signal density strength with the Nis elements software (density per binary area was measured).  The same region of interest (ROI) was selected in the cortex of each animal.  The presence of RFP stained cells was tested in various regions in the brain and in the liver sections. The sections were imaged by fluorescent microscopy (X20, Nikon-TL, or confocal microscope).  We used protocols previously reported by us [33, 35, 36]". We edited the histological studies section in the Manuscript and in the Supplementary materials.

3) Comment: Figure 4: I suggest separating this figure into several independent figures to give enough space to each one and improve its presentation, some images appear distorted. Figure 4A (heat map) needs more size, the protein names are not readable.

Reply: We have separated Fig.4 as the reviewer suggested, and we believe it indeed improved the presentation (now presented in Fig. 4-7). Figure 4 Heatmap has now more size.

4) Comment: Table 2: It is not entirely clear in which condition, AD or treated AD, the increase or decrease occurs.

Reply: We now explained better in the table title: “Comparison of altered proteins due to mitochondria treatment in the liver and in the serum (“increase” or “decrease” in the treated AD relative to the AD mice)”.

5) Comment: Discussion: I suggest addressing in greater depth the different mechanisms of mitochondrial transfer (intercellular and extracellular), as well as the risks and benefits of this novel technique.

Reply: We now addressed in greater depth the mechanism of the physiological process of intercellular exchange of mitochondria, as was reported to take place between neurons and astrocytes. While the mechanism of this exchange is quite clear (via tunneling nanotubes, extracellular vesicles, or cell fusion), the mechanism of transfer of exogenous mitochondria is more complex. While in cell culture, exogenous mitochondria internalization was shown to be mediated by macropinocytosis or endocytosis, the mechanism of internalization of exogenously delivered mitochondrial in vivo  is not  clear. In particular, what is the mechanism of internalization of exogenous mitochondria into the target organ/cells, or is internalization necessarily needed for inducing a beneficial effect in disease animal models.  This is particularly challenging since internalization of exogenous mitochondria into the target organ/cells was not markedly, if at all, evident. Our results showing a beneficial effect on brain without exogenous mitochondria arriving the brain, but rather mediated by the liver via a metabolic liver/brain axis - sheds some light on the mechanism of IV mitochondria transfer therapy.  We believe that  the paragraph in Discussion now starting with: “Transfer of mitochondria is a naturally occurring physiological process….”, explains in depth this issue.

As for the additional important comment of the reviewer regarding the risks and benefits of this novel technique. This therapy shows high efficacy and since collecting mitochondria from various cell types, including an autologous source, is a feasible procedure – it suggests that  the Mitochondria transfer can be easily applied to humans. Indeed, the issue of safety is very important. We tested it in our recent publication of the proof of concept of mitochondrial transfer therapy in the pharmacologically AD-model (ICV injected amyloid-beta mice). Mice treated with mitochondria, as a single dose or as a repeated treatment, showed general health, normal behavior similar to vehicle treated mice, with no macroscopic differences in the internal organs, and no hemorrhages observed in the brains. Moreover, no immune response (tested by the inflammatory marker TNF-a in the serum) was evident at different time points following mitochondrial transfer [33].  This may point to a high safety profile. However, although we did not encounter safety problems, when further developing this therapeutic approach toward clinical trials, safety aspects need to be further addressed.

We now added in the Introduction (in the part describing our recent proof of concept of mitochondrial transfer therapy in the pharmacologically model), the following: "Importantly, this treatment showed high safety profile (normal general health and internal organs, with no immune response)". We also added in the Discussion - Mitochondrial transfer - clinical implications section: "Although we did not encounter any safety problems [33], when further developing this therapeutic approach toward clinical trials, safety aspects need to be further addressed".

Minor concerns:

  1. Comment:Correspondence is indicated with different symbols # and *

Reply: We changed now to *Correspondence.

  1. Comment:The authors have to review and correct several typos throughout the text: e.g. benificial (line571), benfit (line20),…

Reply: Indeed as the reviewer noticed, there are ,unfortunately, some typos. We have corrected these typos, and went through the manuscript to detect and correct additional ones (like “enzymativ” corrected to “enzymatic”).

  1. Comment: Line 23: 5XFAD “mice”?

Reply: We changed now from “model” to “mice”.

  1. Comment: Please avoid abbreviations as much as possible in the abstract so that when it is read it can be understood without having to refer to the full text; or describe the abbreviations in the abstract itself.

Reply: We now describe the abbreviations in the abstract itself: “AD (Alzheimer’s disease)”. We also added List of Abbreviations, according to the suggestion of another reviewer.

  1. Comment: Table 2 appears in the discussion section instead of the results section.

Reply: We have moved the Discussion to start after Table 2.

  1. Comment:  Line 609: …may suggest that some decrease in the production of “A?” is also taking place…( a beta symbol is missing)

Reply: It seems that in the previous PDF version all Ab appeared as A*.

Reviewer 2 Report

In this manuscript, the authors indicate The beneficial effect of mitochondria transfer therapy in 5xFAD mice via liver-serum-brain response. The manuscript can be accepted after addressing the below mentioned corrections.

1.     In introduction section, After sentences, ‘Mitochondrial dysfunction is one of the key pathological processes in the Alzheimer’s-disease (AD) brain.’’ It should be given information. For this purpose the authors can look at the following articles for introduction section: Journal of Molecular Structure 1257, 132613.

2.     After sentences, ‘Due to the involvement of mitochondria impairment in neurodegeneration-related processes, such as oxidative stress…’’ It should be given information. For this purpose the authors can look at the following articles for introduction section: Drug development research 81 (5), 628-636.

3.     After sentences, ‘The pathologies include an influx of plasma proteins into the CNS through the damaged BBB, with fibrinogen in the CNS inducing immune reactions that lead to neurodegeneration.’’ It should be given information. For this purpose the authors can look at the following articles for introduction section: ChemistrySelect 6 (29), 7278-7284.

4.     After sentences, ‘Integrating the changes discovered in the metabolomic and the proteomic profiles of the liver in response to mitochondrial transfer therapy revealed…’’ It should be given information. For this purpose the authors can look at the following articles for introduction section: Journal of Molecular Recognition 36 (3), e3004.

5.     The figures can be further clarified.

6.     There are several English language issues. It should be corrected.

Author Response

Reply to Reviewer 2:

We thank very much the reviewer for his review, providing us comments and suggestions to improve the manuscript, and allowing us to revise our manuscript.  

Please see point to point our answers to the comments:

Reviewer 2

In this manuscript, the authors indicate The beneficial effect of mitochondria transfer therapy in 5xFAD mice via liver-serum-brain response. The manuscript can be accepted after addressing the below mentioned corrections.

  1. Comment:  In introduction section, After sentences, ‘Mitochondrial dysfunction is one of the key pathological processes in the Alzheimer’s-disease (AD) brain.’’ It should be given information. For this purpose the authors can look at the following articles for introduction section: Journal of Molecular Structure 1257, 132613and Arabian Journal of Chemistry 15 (3), 103645

Reply: We indeed agree with the suggestion of the reviewer for the need to expand the information following the sentence ‘Mitochondrial dysfunction is one of the key pathological processes in the Alzheimer’s-disease (AD) brain”. Thank you for suggesting us to look at the interesting articles.

At the beginning of the Introduction, we now wrote: "Alzheimer’s disease (AD) is a neurodegenerative disease affecting many cellular pathways, including protein aggregation, deterioration of neurotransmission, mitochondrial dysfunction, oxidative stress, and neuroinflammation, leading ultimately to neuronal death (Ramachandran 2021 [1] , Anil 2022 [2]). Functional mitochondria are critical for the normal activity of all cells, with neurons highly dependent on mitochondrial function because of   their limited glycolytic capacity,  (Scheffler IE, 2001 [3]). Mitochondria impairment is involved in neurodegeneration-related processes… such as oxidative stress (via reactive oxygen species, ROS, generation – leading to oxidative degeneration of macromolecules [5] and (Demir 2020 [6])….".

  1. Comment:After sentences, ‘Due to the involvement of mitochondria impairment in neurodegeneration-related processes, such as oxidative stress…’’ It should be given information. For this purpose the authors can look at the following articles for introduction section: Drug development research 81 (5), 628-636 and Journal of pharmacy and pharmacology 71 (10), 1576-1583. 

Reply: According to the reviewer recommendation, we added  the suggested article: “Yeliz Demir  (2020).Naphthoquinones, benzoquinones, and anthraquinones: Molecular docking, ADME and inhibition studies on human serum paraoxonase-1 associated with cardiovascular diseases”, (Drug Dev Res 81 (5), 628-636), as a reference to the expanded sentences following our “Due to the involvement of mitochondria impairment in neurodegeneration-related processes, such as oxidative stress [5]’’. We wrote: “…oxidative stress (via reactive oxygen species, ROS, generation – leading to oxidative degeneration of macromolecules  [5] and (Yeliz Demir 2020 [6]).

  1. Comment: After sentences, ‘The pathologies include an influx of plasma proteins into the CNS through the damaged BBB, with fibrinogen in the CNS inducing immune reactions that lead to neurodegeneration.’’ It should be given information. For this purpose the authors can look at the following articles for introduction section: ChemistrySelect 6 (29), 7278-7284 and Archiv der Pharmazie 354 (12), 2100294

Reply: Based on the reviewer suggestion, we now added more details to the sentence: ‘The pathologies include an influx of plasma proteins into the CNS through the damaged BBB, with fibrinogen in the CNS inducing immune reactions that lead to neurodegeneration’’. It is now: "The pathologies include an influx of plasma proteins into the CNS through the damaged BBB, with fibrinogen in the CNS inducing immune reactions (like microglial activation) that lead to neurodegeneration, and also as an active contributor to neurodegenerative diseases such as AD, with altered hemostasis (with Aβ-fibrin(ogen) binding that could thus contribute to Aβ deposition [58]”.  We used this reference which directly focuses in “Fibrinogen and Altered Hemostasis in Alzheimer’s Disease” (Cortes-Canteli, JAD 2012).

  1. Comment: After sentences, ‘Integrating the changes discovered in the metabolomic and the proteomic profiles of the liver in response to mitochondrial transfer therapy revealed…’’ It should be given information. For this purpose the authors can look at the following articles for introduction section: Journal of Molecular Recognition 36 (3), e3004 and Journal of biochemical and molecular toxicology 33 (10), e22381

Reply: We now added in the Results section, following the sentence "‘Integrating the changes discovered in the metabolomic and the proteomic profiles of the liver in response to mitochondrial transfer therapy revealed…’’, the following sentence:  "Similar to the report testing the interaction of lipids with the activity of enzyme-protein in the liver (Palabıyık, 2022[46]).

  1. Comment: The figures can be further clarified.

Reply: We now separated Fig. 4 to Fig. 4-7. We believe that this may clarify the figures, as the reviewer suggests that is needed.

We also mentioned in the legends that the full names of the proteins are presented in the Supplemental Table and excels.

  1. Comment: There are several English language issues. It should be corrected.

Reply: Indeed, as the reviewer noticed, there are, unfortunately, some English corrections needed. We have corrected typos like benificial (line571) to beneficial, benfit (line20) to benefit, like enzymativ to enzymatic corrected to enzymatic.

Reviewer 3 Report

Title: The beneficial effect of mitochondria transfer therapy in 5xFAD mice via liver-serum-brain response

The manuscript offers a lot of data on the efficacy of mitochondrial therapy for the management of Alzheimer's disease. The findings are promising, albeit the authors do point out some limitations about frequency and number of injections. I do have some minor comments.

1.       Please provide Certificate of approval number of the animal study.

2.       Figure 2 must be revised to remove errors.

3.       A list of abbreviations is necessary.

4.       Please explain the rational of using IV tail vein injection at 200 gr/mouse. Furthermore, at least two dosages are usually required to determine efficacy.

5.       Is there any explanation for why mitochondria are only located in the liver? Did you look into other organs?

6.       Further studies/discussion regarding the safety and allergenicity must be determined. Otherwise, section 4.3 may be overstated.

7.       The mechanistic data are complicated. As a result, in order for readers to grasp the core idea of the text, a proposed mechanism of mitochondrial treatment targeting AD must be presented.

Author Response

Reviewer 3

The manuscript offers a lot of data on the efficacy of mitochondrial therapy for the management of Alzheimer      's disease. The findings are promising, albeit the authors do point out some limitations about frequency and number of injections. I do have some minor comments.

  1. Comment: Please provide Certificate of approval number of the animal study.

Reply: Approval number: MD-15-14651-5. We added it in the Methods-Mice section.

  1. Comment:  Figure 2 must be revised to remove errors.

Reply: Indeed, there was errors in copying Figure 2 and transferring to PDF version. We corrected it.

  1. Comment: A list of abbreviations is necessary.

Reply: We have added a List of Abbreviations, after the Acknowledgments.

  1. Comment:  Please explain the rational of using IV tail vein injection at 200 gr/mouse. Furthermore, at least two dosages are usually required to determine efficacy..

Reply: We used in our current study in the 5XFAD mice the 200 mg dose of mitochondria, which we recently showed to have high efficacy in the pharmacologically mouse model of AD. This concentration is in line with the range of the concentrations, 50-150 mg, used in local injections in rat models: such as by intercerebral injection in ischemia model (Huang 2016), into spinal cord after injury (Gollihue 2018), and into the mPFC in Schizophrenia model (Robicsek 2018). Systemic injection of 750 mg mitochondria into the femoral artery in brain ischemic rats was also reported. As such, using 200 mg in our studies in a systemic injection via IV in a smaller animal like a mouse -   seems logical.  These references are cited in the manuscript.  We added to the description of the Transfer protocol in the Methods-Mitochondrial Transfer section: “….  200 mgr mitochondria /mouse (a dose which we used in our recent study in the pharmacological AD model [33]…”

  1. Comment:  Is there any explanation for why mitochondria are only located in the liver? Did you look into other organs?

Reply: As labelled mitochondria injected IV were not detected in the brain, we analyzed their arrival to the liver, the organ which gets the highest blood supply. Moreover, our recent finding that they arrive to the liver in the pharmacological ICV amyloid-beta model, further encouraged us to test the involvement of the liver in response to the mitochondrial transfer in the chronic model of AD, the 5XFAD mice, and study the mechanism. The known phenomenon of cross talk between the liver and the brain in conditions such as hepatic encephalopathy, a brain dysfunction caused by liver insufficiency, provided support to our rational of a liver brain axis in the mitochondrial transfer therapy which showed high efficacy in AD. To explain this point, we added in the discussion: "As an organ which gets the highest blood supply, the accumulation of the IV injected mitochondria in the liver is actually reasonable".

  1. Comment: Further studies/discussion regarding the safety and allergenicity must be determined. Otherwise, section 4.3 may be overstated.

Reply: Indeed, the issue of safety is very important. We tested it in our recent publication of the proof of concept of mitochondrial transfer therapy in the pharmacologically ICV injected amyloid beta mice. Mice treated with mitochondria, as a single dose or as a repeated treatment, showed general health, normal behavior similar to vehicle treated mice, with no macroscopic differences in the internal organs, and no hemorrhages observed in the brains. Moreover, no immune response (tested by the inflammatory marker TNF-a in the serum) was evident at different time points following mitochondrial transfer [33].  This may point to a high safety profile. However, although we did not encounter safety problems, when further developing this therapeutic approach toward clinical trials, safety aspects need to be further addressed.

We now added in the Introduction (in the part describing our recent proof of concept of mitochondrial transfer therapy in the pharmacologically model), the following: "Importantly, this treatment showed high safety profile (normal general health and internal organs, with no immune response)". We also added in section 4.3: "Although we did not encounter any safety problems [33], when further developing this therapeutic approach toward clinical trials, safety aspects need to be further addressed".

  1. Comment: The mechanistic data are complicated. As a result, in order for readers to grasp the core idea of the text, a proposed mechanism of mitochondrial treatment targeting AD must be presented.

Reply: To make it more clear we now added in the Discussion (at the end of the section: A possible liver-serum-brain route in the mitochondria transfer mechanism), the following summarizing paragraph: “Taken together, our proposed mechanism for the beneficial anti-AD effect of IV mitochondrial transfer is that the mitochondria injected into the tail vein arrive (via the heart) to the liver (the organ with the highest blood supply). In the liver they induce a wide metabolic response, leading to a regulated secretion of metabolites into the circulation, which then arrive to the brain via a liver-blood-brain metabolic axis. These metabolites, which are known to have neuroprotective properties, directly or indirectly affect disease pathogenesis, thereby leading to a beneficial anti-AD effect, manifested as changes in cognition, amyloid burden, mitochondrial function, and RNA regulation”. We believe that our Graphical Abstract provides a clear visual summary of this proposed mechanism.

Round 2

Reviewer 1 Report

1) Lines 669-671: there are different font sizes.

2) I suggest including this clarification/explanation in methods/statistics:

"We used the term "trend" aiming to describe results with weak evidence, similar to the definition in VSNI (data science software and experimental design software for biosciences, https://vsni.co.uk/blogs/what-is-a-p-value)"

Author Response

Reply to Reviewer 1 (Minor Revisions):

We thank very much the reviewer for his review and allowing us to revise our manuscript. 

Please see point to point our answers to the comments:

1) Comment: Lines 669-671: there are different font sizes.

Reply: We have now corrected it in the Cells template version (actually in our both previous versions submitted -not in Cells template- there was no different font size).

2) Comment: I suggest including this clarification/explanation in methods/statistics:

"We used the term "trend" aiming to describe results with weak evidence, similar to the definition in VSNI (data science software and experimental design software for biosciences, https://vsni.co.uk/blogs/what-is-a-p-value)"

Reply: We have now added this sentence to the methods/statistics.

Reviewer 2 Report

The manuscript can be accepted this form.

Author Response

Reply to Reviewer 2:

 Thank you for accepting our manuscript for publication.